# Improving Retrieval-Augmented Generation through Multi-Agent Reinforcement Learning

**Yiqun Chen**[1]    **Lingyong Yan**[2]    **Weiwei Sun**[3]    **Xinyu Ma**[2]    **Yi Zhang**[2]
**Shuaiqiang Wang**[2]    **Dawei Yin**[2]    **Yiming Yang**[3]    **Jiaxin Mao**[1]*
[1]Renmin University of China    [2]Baidu Inc.
[3]Carnegie Mellon University
chenyiqun990321@ruc.edu.cn, maojiaxin@gmail.com

## Abstract

Retrieval-augmented generation (RAG) is widely utilized to incorporate external knowledge into large language models, thereby enhancing factuality and reducing hallucinations in question-answering (QA) tasks. A standard RAG pipeline consists of several components, such as query rewriting, document retrieval, document filtering, and answer generation. However, these components are typically optimized separately through supervised fine-tuning, which can lead to misalignments between the objectives of individual components and the overarching aim of generating accurate answers. Although recent efforts have explored using reinforcement learning (RL) to optimize specific RAG components, these approaches often focus on simple pipelines with only two components or do not adequately address the complex interdependencies and collaborative interactions among the modules. To overcome these limitations, we propose treating the complex RAG pipeline with multiple components as a multi-agent cooperative task, in which each component can be regarded as an RL agent. Specifically, we present MMOA-RAG[2], **M**ulti-**M**odule joint **O**ptimization **A**lgorithm for **RAG**, which employs multi-agent reinforcement learning to harmonize all agents' goals toward a unified reward, such as the F1 score of the final answer. Experiments conducted on various QA benchmarks demonstrate that MMOA-RAG effectively boost the overall performance of the pipeline and outperforms existing baselines. Furthermore, comprehensive ablation studies validate the contributions of individual components and demonstrate MMOA-RAG can be adapted to different RAG pipelines and benchmarks.

## 1    Introduction

Large Language Models (LLMs) have been widely applied to tasks such as question answering [1, 23], information retrieval [45, 2], various forms of reasoning [15, 13], and evaluation [11, 8]. However, since LLMs cannot promptly update their internal knowledge after pre-training, they are still prone to generating outdated or fabricated responses [58]. To address these challenges, Retrieval-Augmented Generation (RAG) enhances the generative capabilities of LLMs by retrieving relevant information from external knowledge sources. Recent RAG systems are often built as complex pipelines comprising multiple interconnected modules [10], including query rewriting [30, 19], first-stage retrieval [25, 40], re-ranking [36, 37], document preprocessing [22, 27], and answer generation [40, 44].

The complexity of RAG systems makes their optimization particularly challenging. Standard supervised fine-tuning (SFT) optimizes each module independently using human-annotated data. However,

---

*Corresponding author.

[2]The code of MMOA-RAG is on `https://github.com/chenyiqun/MMOA-RAG`.

this often results in misalignment between the objectives of individual components and the overarching goal of the system of generating high-quality results. For example, retrieval modules are frequently trained on human-labeled relevance data to optimize metrics such as nDCG[18]. However, this process does not address the disconnect between document relevance and response quality - documents with high relevance scores do not always contribute to generating accurate answers [6].

To address this issue, existing work on end-to-end optimization for RAG, such as [24, 12, 25, 37] aims to propagate rewards from the final output to intermediate modules using techniques such as attention distributions [17], generation probability [25, 56], and expectation maximization (EM) iterations [42, 36]. However, earlier approaches primarily focus on simplified pipelines with only two components-a retriever and a generator-and fail to provide a generalizable framework for jointly optimizing complex systems with multiple components and richer interdependencies. More recent methods attempt to eliminate the need for module-specific rewards by leveraging algorithms like Direct Preference Optimization (DPO) [33] and Proximal Policy Optimization (PPO)[39]. Nonetheless, these methods still concentrate on optimizing individual RAG modules in isolation, without adequately modeling the collaborative dynamics between interacting components [27, 30, 22]. Effectively capturing interdependencies among multiple modules and jointly optimizing complex RAG architectures remains an open research challenge.

In this paper, we propose a novel approach called the **M**ulti-**M**odule joint **O**ptimization **A**lgorithm (MMOA-RAG) to enable joint optimization across multiple modules in a RAG system. Our framework treats each intermediate component in the RAG pipeline as an agent and formulates the optimization process as a **Co**operative **M**ulti-**A**gent **R**einforcement **L**earning (Co-MARL) problem, where the agents (i.e., modules) work together to maximize a shared reward for the final outcome.

Specifically, we focus on applying MMOA-RAG to a RAG pipeline that includes four key modules: a query rewriter, a fixed document retriever, a document selector, and an answer generator. Our primary objective is to optimize these modules by defining the final reward as the correctness of the generated response, measured by the F1 score against the ground-truth answer. To achieve this, we leverage the Multi-Agent PPO (MAPPO) algorithm [55], which facilitates collaborative optimization in a fully cooperative setting. This means that all modules work in a cooperative way, with their optimization goals aligned toward producing high-quality answers. Unlike previous methods that rely on DPO [62, 61] or PPO [30, 22], MMOA-RAG offers greater flexibility for different pipeline designs and excels at promoting collaboration among multiple modules. This end-to-end optimization ensures that each module's objectives are consistently aligned with the overarching goal of generating accurate responses.

To demonstrate the effectiveness of the MMOA-RAG modeling and optimization approach, we conducted experiments on three publicly available QA datasets, HotpotQA [53], 2WikiMultihopQA [14] and AmbigQA [31], based on Llama-3-8B-Instruct [7]. The experimental results indicate that MMOA-RAG achieves better performance than a series of existing optimization methods for RAG. Additionally, we performed extensive ablation studies to investigate the effectiveness and advantages of jointly optimizing mulitple modules in the RAG system and the generalizability of MMOA-RAG across different RAG pipelines.

Our main contributions are as follows:

- We innovatively model RAG as a multi-agent collaborative task, treating multiple modules within the RAG pipeline as individual agents.
- We employ a multi-agent reinforcement learning algorithm to jointly optimize a sophisticated RAG system with four key modules: a query writer, a fixed document retriever, a document selector, and an answer generator.
- We conduct extensive experiments to verify and demonstrate the effectiveness and generalizability of the proposed framework.

## 2 Related Works

**End-to-end Optimization in OpenQA** Lewis et al. [25] introduces Retrieval-Augmented Generation as RAG, which combines pre-trained language models with non-parametric memory for improved performance on knowledge-intensive NLP tasks. And some other methods [24, 12, 17, 56] proposed end-to-end optimizing framework for OpenQA system.

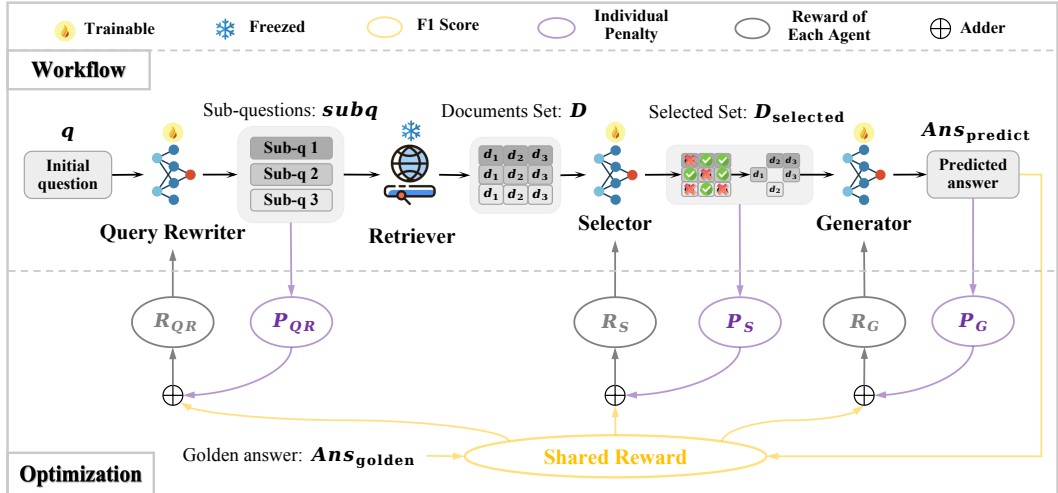

Figure 1: The overall framework of MMOA-RAG.

**RAG without Parameters Updating** Some works [20, 50, 44, 46] design a novel RAG mechanism without parameters updating, enhancing the performance of LLMs on question answering tasks.

**RAG with Parameters Updating** Some works [51, 57, 48] optimize RAG with supervised fine-tuning. PPO [39] is used by some works [39, 30, 22, 9, 19] to fine-tune LLMs. Specifically, Search-r1 [21] and R1-Searcher [43] both use answer-based reward to improve the reasoning in RAG. Additionally, DPO [33] or similar alignment algorithms are used to optimize LLMs in RAG task [33, 62, 61, 59, 27]. More detailed related works can be seen in Appendix A.

## 3 Method

### 3.1 Modeling RAG as Co-MARL

In this work, we conceptualize the RAG procedure within a cooperative multi-agent reinforcement learning (Co-MARL) framework. Within this framework, each module of the RAG pipeline functions as an individual RL agent. The overarching objective of this multi-agent system is to produce high-quality answers, which aligns with the individual goals of each module.

We define the tuple $\langle \mathcal{G}, \mathcal{O}, \mathcal{A}, \mathcal{R} \rangle$, where $\mathcal{G}$ denotes the set of agents in the Co-MARL system, $\mathcal{O}$ represents the observation information available to each agent, $\mathcal{A}$ constitutes the action space accessible to each agent, and $\mathcal{R}$ is the reward shared among all agents. The ultimate aim is to maximize this shared reward, thereby achieving higher evaluation metrics and enhancing the overall performance of the RAG system.

In this paper we utilizes Multi-Agent PPO (MAPPO) [55], which is an extension of the PPO algorithm [39] for multi-agent environments, to optimize the policy for each agent in the Co-MARL framework. In fully cooperative settings, unlike PPO, which focuses on single-agent scenarios with individual reward, MAPPO employs a shared global reward to promote cooperation among all agents.

### 3.2 Overall of MMOA-RAG

RAG systems typically follow a modular architecture composed of multiple interconnected components. Figure 1 illustrates the architecture of our MMOA-RAG framework, which consists of four primary modules: the Query Rewriter, the Retriever, the Selector, and the Generator:

- **Query Rewriter** reformulates the initial question $q$, which may be too complex or ambiguous to resolve with a single retrieval, into a set of sub-questions denoted as $subq$.

- **Retriever** retrieves relevant documents from the corpus for each sub-questions, respectively, and outputs a set of candidate document $D$.

- **Selector** further filters $D$ to obtain a subset of documents $D_{\text{selected}}$ that is useful for generating the final answer to the initial query $q$.

- **Generator** leverages $D_{\text{selected}}$ to generate the predicted answer $Ans_{\text{predict}}$ to the initial question.

Since the Query Rewriter, Selector, and Generator modules can all be implemented using LLMs, they can be treated as RL agents [32], enabling parameter updates through reward signals. To optimize computational efficiency, these three modules can share the same LLM. Additionally, given the difficulty of modeling the Retriever module as an RL agent, we use a fixed Retriever and regard it as a part of the environment [3].

The focus of the MMOA-RAG framework is on the collaborative optimization of multiple modules to align their individual objectives with the ultimate goal of generating high-quality answers. We use metrics from the Generator's predicted answer $Ans_{\text{predict}}$, such as F1 score, as a shared reward $R_{\text{shared}}$. Given the fully cooperative nature of the modules in the RAG system, $R_{\text{shared}}$ can be used to train all agents, a common approach in existing MARL literature [35, 55, 4]. Additionally, to ensure training stability and accelerate convergence in the multi-agent system, we design penalty terms $P_{QR}$, $P_S$, and $P_G$ for each agent. A more detailed explanation will be provided in Section 3.3.

### 3.3 Detailed Configuration for Each Agent

In this section, we will provide a detailed explanation of each element in the tuple $\langle \mathcal{G}, \mathcal{O}, \mathcal{A}, \mathcal{R} \rangle$ mentioned in Section 3.1. Here, $\mathcal{G} = \{$Query Rewriter (QR), Selector (S), Generator (G)$\}$ represents all agents. In the following, we introduce the essential elements for each agent $i \in \mathcal{G}$: the observation information $O_i \in \mathcal{O}$, the action space $A_i \in \mathcal{A}$, and the reward function $R_i$.

#### 3.3.1 Elements of Query Rewriter

**Observation** of Query Rewriter is defined as Equation (1), which contains prompt of Query Rewriter $Prompt_{QR}$ (as shown in Table 3) and the initial question $q$.

$$O_{QR} = \{Prompt_{QR}, q\} \tag{1}$$

**Action Space** of Query Rewriter corresponds to the vocabulary of LLMs, denoted as $\mathcal{V}$, as we prompt the LLM to generate one or more sub-questions based on $q$.

$$A_{QR} = \mathcal{V} \tag{2}$$

**Reward Function** of the Query Rewriter is defined as shown in Equation (3). Here, $R_{\text{shared}}$ can be the metric for the final answer, depicted as the yellow section in Figure 1. In this paper, we utilize the F1 score of the predicted answer, $Ans_{\text{predict}}$, as the shared reward. The term $P_{QR}$ serves as a penalty to discourage the Query Rewriter from generating an excessive number of sub-questions during training. Specifically, $P_{QR}$ is assigned a value of -0.5 if the number of sub-questions exceeds four, and it is set to 0 if the number of sub-questions is four or fewer.

$$R_{QR} = R_{\text{shared}} + P_{QR} \tag{3}$$

#### 3.3.2 Elements of Selector

**Observation** of Selector is defined as Equation (4), which contains prompt of Selector $Prompt_S$ (as shown in Table 4), the initial question $q$ and the candidate documents set $D$ with $K$ documents.

$$O_S = \{Prompt_S, q, D\} \tag{4}$$

**Action Space** of Selector only comprises of several words as Equation (5). Since the function of the Selector is to output the IDs of candidate documents helpful to answering the initial question $q$, the action space is constrained to this limited set of words. This constraint can significantly reduce the exploration space of the Selector and provide a more stable training process.

$$A_S = \{\text{"0", "1", ..., "K-1", "Document", ","}\} \tag{5}$$

---

[3]Recent studies in generative IR (see [26] for a survey) have explored using generative models for retrieval. But we choose a more traditional dense retrieval model [16] as the first-stage retriever and leave the optimization of the first-stage retriever for future work.

**Reward Function** of Selector also contains two terms, which are $R_{\text{shared}}$ and $P_S$. And $P_S$ is a penalty term designed to prevent the Selector from generating duplicate document IDs and from outputting IDs that do not conform to the specified format (e.g., Document0,Document3,Document9). When the Selector outputs duplicate document IDs or fails to adhere to the specified format, $P_S$ is set to -1; otherwise, $P_S$ is set to 0.

$$R_S = R_{\text{shared}} + P_S \tag{6}$$

### 3.3.3 Elements of Generator

**Observation** of Generator is in Equation (7), containing prompt of Generator $Prompt_G$ (as shown in Table 5), the initial question $q$ and the selected candidate documents set $D_{\text{selected}}$ given by Selector.

$$O_G = \{Prompt_G, q, D_{\text{selected}}\} \tag{7}$$

**Action Space** of Generator $A_G$ is the same as Query Rewriter.

$$A_G = A_{QR} = \mathcal{V} \tag{8}$$

**Reward Function** of Generator contains $R_{\text{shared}}$ and penalty term $P_G$, which is used to constrain the model from generating excessively long content. When the generated answer exceeds a certain length, $P_G$ is set to -0.5; otherwise, it is set to 0. In fact, the values of each penalty $P_i$ ($i \in \mathcal{G}$) are mostly 0, and they only become negative when the output does not meet the requirements.

$$R_G = R_{\text{shared}} + P_G \tag{9}$$

## 3.4 Training Process of MMOA-RAG

### 3.4.1 Warm Start with SFT

In preparation for joint optimization of multiple modules using Multi-Agent PPO, it is essential to perform a warm start for each trainable module. The warm start enables the model to better adhere to instructions across diverse tasks and reduces the exploration space during MARL joint training, thereby enhancing the efficiency of exploration and exploitation.

Within the MMOA-RAG framework, there are three trainable modules: the Query Rewriter, the Selector, and the Generator. Consequently, we construct the training data for the SFT of each corresponding task and perform the SFT to get the warm-up checkpoints for each trainable modules. The details of constructing training data can be seen in Appendix B.

### 3.4.2 Multi-Agent Optimization

After undergoing SFT, the LLM demonstrates an improved ability to follow instructions while executing the functions of Query Rewriter, Selector, and Generator. The RAG system also achieves relatively satisfactory warm-start performance. To further enhance the performance of the RAG system, which is modeled as a fully cooperative multi-agent system, it is crucial to conduct joint training of multiple agents to strengthen collaboration among them.

We adopt a setup similar to Multi-Agent PPO [55] in Starcraft II, where multiple agents share a global reward, optimizing $\mathcal{G} = \{QR, S, G\}$ with $R_{\text{shared}}$. To reduce computational overhead, we apply the parameter-sharing mechanism among agents, allowing QR, S, and G to utilize the same LLM.

In the multi-agent optimization process, there are three models to consider: the Actor model, the Critic model, and the SFT model. The parameters for these models are denoted as $\theta$, $\phi$, and $\theta_{\text{SFT}}$, respectively. The role of the Actor model is to provide the response $Answer_i$ based on the observation $O_i$ for each agent $i$. The Critic model is responsible for estimating the state-value function $V_\phi^{i,t}$, which is a classic setup in Actor-Critic architecture within RL algorithms. The SFT model serves as a baseline for the Actor model, similar to InstructGPT [32]. The objective is to update the parameters of both the Actor and Critic models. The overall loss function, $\mathcal{L}(\theta, \phi)$, consists of two terms: $\mathcal{L}_{\text{Actor}}(\theta)$ and $\mathcal{L}_{\text{Critic}}(\phi)$:

$$\mathcal{L}(\theta, \phi) = \mathcal{L}_{\text{Actor}}(\theta) + \alpha * \mathcal{L}_{\text{Critic}}(\phi) \tag{10}$$

The Actor loss function presented in Equation (11) is similar to that used in the typical single-agent PPO [39] algorithm. The primary difference is that multiple agents are being optimized. In Equation

(11), $i \in \mathcal{G}$ denotes the three agents: Query Rewriter, Selector, and Generator. The term $r_t^i$ in Equation (12) denotes the importance sampling ratio, which measures the difference between the new and old policies. The expression $\hat{A}_{\pi_\theta}^{i,t}$ in Equation (13) is the advantage function, estimated using Generalized Advantage Estimation (GAE) [38]. The variable $\delta_t^i$ in Equation (14) is known as the temporal difference (TD) error at time step $t$.

$$\mathcal{L}_{\text{Actor}}(\theta) = \sum_i \sum_t \min\left(r_t^i \hat{A}_{\pi_\theta}^{i,t}, \ \text{clip}\left(r_t^i, 1-\epsilon, 1+\epsilon\right) \hat{A}_{\pi_\theta}^{i,t}\right) \tag{11}$$

$$r_t^i = \frac{\pi_\theta(a_t^i \mid s_t^i)}{\pi_{\theta_{\text{old}}}(a_t^i \mid s_t^i)} \tag{12}$$

$$\hat{A}_{\pi_\theta}^{i,t} = \sum_{l=0}^{\infty} (\gamma\lambda)^l \delta_{t+l}^i \tag{13}$$

$$\delta_t^i = R(s_t^i, a_t^i) + \gamma V_\phi(s_{t+1}^i) - V_\phi(s_t^i) \tag{14}$$

Similar to InstructGPT [32], the final reward function $R(s_t^i, a_t^i)$ is defined in Equation (15). The distinction is that our approach does not require a trained reward model, as we use the evaluation metric (F1 score) of the predicted answers $Ans_{\text{predict}}$ of Generator as the shared reward $R_{\text{shared}}$ for all agents. The penalty term $P_i$ can also be easily obtained from the output of each agent, as introduced in Section 3.3. The components $R_i$ in Equation (15) are defined in Equations (3), (6), or (9). And $Answer_i$ represents the output generated by each agent $i$ based on its individual observation $O_i$.

$$R(s_t^i, a_t^i) = \begin{cases} 0, & \text{if } t < T \\ \underbrace{R_{\text{shared}} + P_i}_{R_i, \text{ and } i \in \mathcal{G}} -\beta \log\left(\frac{\pi_\theta(Answer_i|O_i)}{\pi_{\theta_{\text{SFT}}}(Answer_i|O_i)}\right), & \text{if } t = T \end{cases} \tag{15}$$

The loss function of the Critic model, as shown in Equation (16), employs a clipping operation similar to the Actor model. Here, $\Delta V_{i,t} = V_\phi^{i,t} - V_{\text{target}}^{i,t}$, where $V_\phi^{i,t} = V_\phi(s_t^i)$. The term $V_{\text{target}}^{i,t}$ represents the cumulative return and $s_t^i$ is the state-value function.

$$\mathcal{L}_{\text{Critic}}(\phi) = \sum_i \sum_t \max\left[(\Delta V_{i,t})^2, \left(\text{clip}\left(V_\phi^{i,t}, V_{\phi_{\text{old}}}^{i,t} \pm \epsilon\right) - V_{\text{target}}^{i,t}\right)^2\right] \tag{16}$$

The pseudocode for multi-agent optimization based on MAPPO is shown in **Algorithm 1** in Appendix C, which corresponds to the overall framework of MMOA-RAG depicted in Figure 1. For a specific question, the first step is to execute the `Collect Rollout` process. This process involves passing through the Query Rewriter, Retriever, Selector, and Generator, and the computed tuple $\mathcal{T} = ((O_{QR}, subq, R_{QR}), (O_S, IDs, R_S), (O_G, Ans_{\text{predict}}, R_G))$ is stored in the replay buffer $\mathcal{M}$. Next, the `Policy and Value Optimization` process is executed where the GAE is used to estimate the advantage function $\hat{A}_{\pi_\theta}^{i,t}$. Subsequently, the overall loss function $\mathcal{L}(\theta, \phi)$ is calculated, and the parameters of both the Actor and Critic models are updated. Additionally, to accelerate the entire training process, we can run a minibatch in parallel. Ultimately, we obtain a well-trained Actor model used for subsequent inference and evaluation.

## 4 Experiments

Our experiments mainly aim to explore the following research questions:

**RQ1:** How does our MMOA-RAG perform compared to existing RAG optimization methods?

**RQ2:** How does the joint optimization of individual modules in the RAG pipeline contribute to the effectiveness of MMOA-RAG framework?

**RQ3:** Can MMOA-RAG exhibit generalizability across different RAG systems?

### 4.1 Experimental Settings

**Datasets and Evaluation** We conducted experiments using MMOA-RAG alongside various baseline models across three open-domain QA datasets: HotpotQA [53], 2WikiMultihopQA [14], and AmbigQA [31]. The candidate documents are all retrieved from Wikipedia passages for three datasets.

Table 1: Performance for different methods across datasets. All the results in this table are obtained using Contriever [16] as the retrieval model. In each dataset, the highest baseline value is underscored. The symbol $\Delta$ displays the improvement of MMOA-RAG over the best baseline.

| Methods | HotpotQA | | | 2WikiMultihopQA | | | AmbigQA | | |
|---|---|---|---|---|---|---|---|---|---|
| | Acc | EM | F1 | Acc | EM | F1 | Acc | EM | F1 |
| LLM w/o RAG | 25.08 | 21.31 | 31.18 | 27.78 | 23.68 | 29.47 | 27.21 | 20.96 | 33.42 |
| Vanilla RAG w/o train | 27.99 | 20.62 | 30.67 | 31.94 | 13.91 | 22.84 | 31.09 | 22.42 | 33.56 |
| Vanilla RAG w SFT | 36.18 | 32.30 | 44.49 | 39.47 | 38.28 | 43.36 | 34.41 | 30.74 | 44.36 |
| SELF-RAG [1] | 30.42 | 27.77 | 38.93 | 36.32 | 35.39 | 38.86 | 28.35 | 25.70 | 39.04 |
| RetRobust [54] | 37.69 | 34.60 | 46.49 | 41.02 | 39.73 | 44.51 | 35.13 | 32.37 | 44.78 |
| Rewrite-Retrieve-Read [30] | 38.03 | 33.93 | 46.32 | 40.40 | 39.17 | 44.17 | 35.94 | 31.90 | 45.92 |
| BGM [22] | 36.05 | 32.76 | 44.54 | 39.61 | 38.61 | 43.29 | 36.01 | 32.53 | 45.76 |
| RAG-DDR [27] | 35.20 | 32.65 | 44.26 | 40.49 | 39.45 | 44.18 | 36.25 | 32.55 | 45.83 |
| MMOA-RAG (ours) | **39.15** | **36.15** | **48.29** | **42.73** | **41.52** | **46.40** | **38.85** | **34.75** | **48.59** |
| $\Delta$ | **+1.12** | **+1.55** | **+1.80** | **+1.71** | **+1.79** | **+1.89** | **+2.60** | **+2.20** | **+2.67** |

We employ three key evaluation metrics—Accuracy, Exact Match (EM), and F1 score—to assess the performance of the RAG methods.

**Implementation Details** We utilize Contriever [16] as the Retriever for most experiments. And the Selector consistently receives a fixed set of $K = 10$ documents as input. Besides, we employ Llama-3-8B-Instruct [7] as the foundational LLM for all the baselines and MMOA-RAG.

We compare MMOA-RAG with different baseline methods: **LLM w/o RAG**, **Vanilla RAG w/o train**, **Vanilla RAG w SFT**, **SELF-RAG** [1], **RetRobust** [54], **Rewrite-Retrieve-Read** [30], **BGM** [22], **RAG-DDR** [27].

The detailed introduction of experimental settings and baselines can be seen in Appendix D.

## 4.2 Comparisons with Other Methods

We conducted a comparative analysis of MMOA-RAG against multiple baselines, with the results presented in Table 1. To ensure fairness in comparison, all methods utilized Llama-3-8B-Instruct as the backbone LLM, and all baselines were re-implemented according to the settings delineated in Appendix D.2.

Firstly, as shown in Table 1, MMOA-RAG demonstrates superior performance across all metrics and datasets, highlighting its effectiveness. Additionally, it is noteworthy that Vanilla RAG w/o train achieves comparable results to LLM w/o RAG across various metrics. This observation suggests that the pre-trained Llama-3-8B-Instruct struggles to effectively leverage external knowledge for answer generation, likely due to the absence of RAG-related tasks in its pre-training process, which limits its external knowledge utilization. In contrast, Vanilla RAG w SFT exhibits substantial improvements over Vanilla RAG w/o train across all evaluation metrics. This indicates that the SFT-enhanced Llama-3-8B-Instruct is adept at utilizing external knowledge, successfully extracting valuable information from noisy candidate documents to enhance the quality of generated answers.

The Rewrite-Retrieve-Read and BGM approaches enhance Vanilla RAG by respectively integrating a query rewrite module and a bridge module, each of which is trained using the PPO algorithm. As indicated in Table 1, on the multi-hop datasets HotpotQA and 2WikiMultihopQA, Rewrite-Retrieve-Read surpasses BGM, suggesting that the inclusion of a query rewrite module is more effective than adding a bridge module for these multi-hop datasets. Conversely, on the single-hop dataset AmbigQA, the performance of Rewrite-Retrieve-Read and BGM is relatively similar. Our MMOA-RAG can be conceptualized as augmenting Vanilla RAG by integrating both a Query Rewriter and a Selector, whose roles are akin to the query rewrite module in Rewrite-Retrieve-Read and the bridge module in BGM. The primary advantage of MMOA-RAG lies in its simultaneous optimization of the Query Rewriter, Selector, and Generator modules. This is achieved by aligning the objectives of these modules with the goal of generating higher-quality answers via MAPPO. The experimental results presented in Table 1 further illustrate that MMOA-RAG significantly outperforms Rewrite-Retrieve-Read, BGM, and other baselines.

In addition, we tested various methods based on other retriever, BGE [49] and E5 [47], and the results can be seen in Table 7 in Appendix E. And we also perform out-of-domain experiments, the results can be seen in Table 8 in Appendix F. In summary, The results in Table 1, Table 7, Table 8, and the

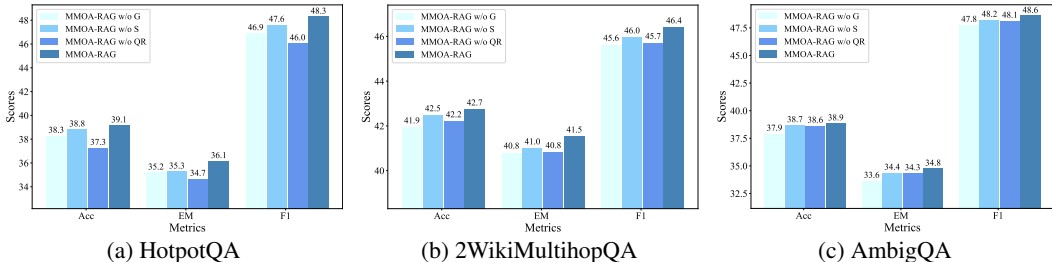

|  (a) HotpotQA | (b) 2WikiMultihopQA | (c) AmbigQA |

Figure 2: Ablation about Optimizing Different Agents. In this figure, MMOA-RAG w/o $i$ ($i \in \{QR, S, G\}$) denote the variant where agent $i$ is excluded from the complete optimization process of multi-agent joint optimization.

analysis in this Section 4.2 jointly answer the **RQ1**. And the case study can be found in Appendix I, from which we can intuitively understand the advantages of multi-module joint optimization.

### 4.3 Ablation Experiments on the Optimization of Different Agents

To demonstrate the necessity of multi-agent joint optimization in RAG systems, we present ablation experiments in this section. The MMOA-RAG framework, depicted in Figure 1, consists of three agents: $i \in \{$Query Rewriter (QR), Selector (S), Generator (G)$\}$. In Figure 2, MMOA-RAG w/o $i$ denotes the variant where agent $i$ is excluded from the complete optimization process of multi-agent joint optimization.

As illustrated in Figure 2, the complete version of MMOA-RAG, where all three modules are jointly optimized, delivers the highest performance. This underscores the effectiveness of multi-agent joint optimization within the RAG system and validates the importance of optimizing multiple modules concurrently. Additionally, the MMOA-RAG w/o S variant achieves the best performance among the three ablation configurations. The Selector's primary function is to refine the candidate document set $D$, yielding a higher-quality subset $D_{\text{selected}}$, which enhances the Generator's ability to produce a superior answer $Ans_{\text{predict}}$. However, through the joint optimization by MAPPO, the Generator acquires some denoising capabilities. Consequently, satisfactory results can be achieved even when the Selector is not optimized during joint optimization.

We also present the trajectory of the shared reward $R_{\text{shared}}$ during the training process based on abla-

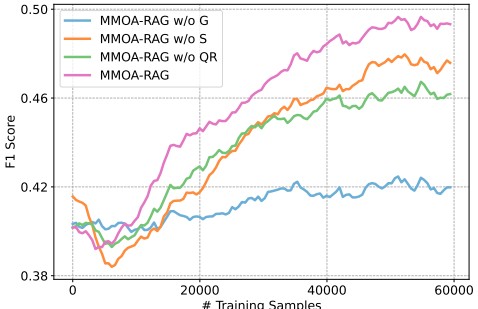

Figure 3: Ablation experiments on AmbigQA dataset. The horizontal axis represents the number of training samples, while the vertical axis denotes the shared reward $R_{\text{shared}}$ (F1 score) during the training process.

tion experiments conducted on the AmbigQA dataset, as shown in Figure 3. From Figure 3, it is evident that the reward curve for MMOA-RAG demonstrates the fastest convergence rate and achieves the highest final convergence value. This underscores the effectiveness of joint optimization across multiple modules in significantly and efficiently enhancing the performance of the RAG system. Furthermore, the training curve for MMOA-RAG w/o G in Figure 3 is noticeably slower compared to other algorithms, and the test results for MMOA-RAG w/o G on the AmbigQA dataset, as shown in Figure 2, are the poorest. These findings suggest that the Generator module is the most critical component for the single-hop AmbigQA dataset.

The results in Figure 2 and Figure 3 answer the **RQ.2** that it is more effective to optimize multiple modules in a RAG system simultaneously.

### 4.4 Generality Experiments on RAG Systems with Varying Module Configurations

In this section, we evaluate the performance of MMOA-RAG in optimizing RAG systems with different numbers of agents, as detailed in Table 2. In Table 2, QR+S+G represents the RAG

Table 2: Generality Experiments on RAG Systems with Varying Module Configurations. In the second column, SFT and MAPPO refer to the current module configuration following the warm start training stage (Section 3.4.1) and the MAPPO joint training stage (Section 3.4.2), respectively. The symbol $\Delta$ signifies the enhancement achieved in the MAPPO stage relative to the SFT stage.

| Modules | Training Stage & Delta | HotpotQA | | | 2WikiMultihopQA | | | AmbigQA | | |
|---|---|---|---|---|---|---|---|---|---|---|
| | | Acc | EM | F1 | Acc | EM | F1 | Acc | EM | F1 |
| QR+S+G | SFT | 36.00 | 33.04 | 44.69 | 39.54 | 38.50 | 42.97 | 36.55 | 32.60 | 46.71 |
| | MAPPO | 39.15 | 36.15 | 48.29 | 42.73 | 41.52 | 46.40 | 38.85 | 34.75 | 48.59 |
| | $\Delta$ | **+3.15** | **+3.11** | **+3.60** | **+3.19** | **+3.02** | **+3.43** | **+2.30** | **+2.15** | **+1.88** |
| S+G | SFT | 34.25 | 32.18 | 43.14 | 38.93 | 37.97 | 42.40 | 35.85 | 32.35 | 45.82 |
| | MAPPO | 38.23 | 34.85 | 47.07 | 41.79 | 40.57 | 45.25 | 37.60 | 33.90 | 47.19 |
| | $\Delta$ | **+3.98** | **+2.67** | **+3.93** | **+2.86** | **+2.60** | **+2.85** | **+1.75** | **+1.55** | **+1.37** |
| QR+G | SFT | 36.76 | 32.78 | 45.00 | 39.15 | 37.89 | 42.91 | 35.50 | 31.50 | 45.31 |
| | MAPPO | 38.90 | 35.89 | 47.94 | 42.43 | 41.01 | 46.19 | 37.65 | 33.50 | 47.53 |
| | $\Delta$ | **+2.14** | **+3.11** | **+2.94** | **+3.28** | **+3.12** | **+3.28** | **+2.15** | **+2.00** | **+2.22** |

framework depicted in Figure 1, illustrating a multi-agent system composed of three agent, Query Rewriter, Selector and Generator. The configuration S+G results from omitting the Query Rewriter agent, relying solely on the initial question $q$ for retrieval, thereby configuring the RAG system as a two-agent (Selector and Generator) system. Conversely, QR+G denotes the exclusion of the Selector agent, forming a RAG pipeline consisting of two agents, Query Rewriter and Generator. The second column of Table 2 specifies that SFT refers to the warm start of all agents in the corresponding RAG system through supervised fine-tuning, while MAPPO refers to the joint optimization of all agents built upon SFT utilizing the MAPPO framework. The notation $\Delta$ is used to denote the performance enhancement achieved by MAPPO compared to SFT.

The experimental results in Table 2 reveal that RAG systems optimized using joint MAPPO consistently outperform those using only SFT across all datasets. This finding underscores the robust generalizability of the MMOA-RAG joint optimization approach, yielding significant performance improvements across diverse RAG configurations. Notably, the performance gains from MAPPO over SFT are approximately three percentage points on multi-hop datasets such as HotpotQA and 2WikiMultihopQA, while improvements on the single-hop dataset AmbigQA are around two percentage points. This difference may stem from the greater complexity inherent to multi-hop datasets, which potentially exacerbates misalignment among different modules during the SFT stage. These results further highlight the necessity of multi-module joint optimization, especially in the context of more challenging multi-hop datasets.

The results presented in Table 2 demonstrate the effectiveness of MMOA-RAG in optimizing various RAG systems across different configurations, thereby answering **RQ.3**.

## 5   Conclusions and Future Works

In this paper, we model the RAG system as a multi-agent collaborative task, wherein we consider the Query Rewriter, Selector, and Generator modules as learnable RL agents. We employ a multi-agent reinforcement learning algorithm to jointly optimize these agents, aligning the optimization goals of multiple modules with the ultimate objective of generating high-quality answers.

Our experiments demonstrate the effectiveness of our modeling approach and joint optimization method. Comprehensive ablation studies confirm the necessity and generality of multi-module joint optimization, establishing MMOA-RAG as an effective approach for optimizing RAG systems.

In future works, we intend to explore the application of MMOA-RAG in more complex workflows. This exploration will include scenarios where the RAG workflows are organized as a directed acyclic graph, as well as situations involving dynamic workflows within agentic RAG. Additionally, it is important to assess the cost and latency associated with specific modules within the RAG system. In this regard, the design of the reward function should not be exclusively based on evaluation metrics, such as the F1 score highlighted in this paper. Instead, it should aim to strike a balance between effectiveness and cost in those partially cooperative RAG scenarios.

## Acknowledgements

This research was supported by the Natural Science Foundation of China (61902209, 62377044), Intelligent Social Governance Platform, Major Innovation & Planning Interdisciplinary Platform for the "Double-First Class" Initiative, Renmin University of China, the Fundamental Research Funds for the Central Universities, the Research Funds of Renmin University of China (22XNKJ15), and Beijing Nova Program.

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

# Appendix

## A  Detailed Related Works

### A.1  End-to-end Optimization in OpenQA

ORQA [24] is an open-domain QA system that learns end-to-end evidence retrieval and answer generation using only question-answer pairs, enabled by pretraining with an Inverse Cloze Task. REALM [12] is an end-to-end optimizing framework that enhances language model pre-training with a retrieval-augmented approach. [25] introduces Retrieval-Augmented Generation as RAG, a model that combines pre-trained language models with non-parametric memory for improved performance on knowledge-intensive NLP tasks. [17] propose a knowledge distillation method to train retriever models using synthetic labels derived from reader model attention scores. Stochastic RAG [56] introduces a novel end-to-end optimization framework for RAG through expected utility maximization.

### A.2  RAG without Parameters Updating

These methods typically involve designing a novel RAG mechanism to enhance the performance of LLMs on question answering tasks. For example, DSP [23] leverages sophisticated interactions between retrieval and language models to address knowledge-intensive NLP tasks. FLARE [20], an active retrieval augmented generation method, enhances text generation by dynamically retrieving relevant information throughout the process, showing superior performance across various long-form knowledge-intensive tasks. ITER-RETGEN [40] is an iterative retrieval-generation synergy method that enhances retrieval-augmented large language models by synergistically combining retrieval and generation in an iterative manner. Search-in-the-Chain [50] is a framework that interactively enhances Large Language Models with search capabilities to improve performance on complex, knowledge-intensive tasks. SELF-RAG [1] enhances language model quality and factuality through self-reflective retrieval and generation. DRAGIN [44] is a dynamic RAG framework that addresses the real-time information needs of LLMs during text generation, enhancing performance on knowledge-intensive tasks. GenGround [41] synergizes large language model knowledge with external documents to enhance multi-hop question answering through an iterative process of generating answers and grounding them in evidence. Astute RAG [46] is a approach that enhances the robustness of Retrieval-Augmented Generation for Large Language Models by adaptively integrating internal and external knowledge while resolving knowledge conflicts.

### A.3  RAG with Parameters Updating

#### A.3.1  Optimizing RAG with SFT

INFO-RAG [51], an unsupervised training method, enhances the capacity of large language models to integrate and refine information from retrieved texts. LongRAG [57] introduces a dual-perspective retrieval-augmented generation system to enhance understanding of complex long-context knowledge for improved performance in long-context question answering tasks. In INSTRUCTRAG [48], generation accuracy and trustworthiness are enhanced by explicitly denoising retrieved information through self-synthesized rationales, outperforming standard RAG approaches without additional supervision.

#### A.3.2  Optimizing RAG with RL

Some existing works use PPO [39] algorithm to fine-tune LLMs. In Rewrite-Retrieve-Read framework [30], a small language model is trained with reinforcement learning to rewrite queries for RAG. BGM [22] proposes a novel bridge mechanism between retrieval model and LLMs and uses PPO to optimize the parameters of the bridge to filter for more helpful documents. SMARTRAG [9] optimizes an iterative RAG framework with reward, which includes a decision maker and a policy network. RAG-Star [19] is a reasoning approach that combines Monte Carlo Tree Search (MCTS) to improve the complex reasoning abilities of LLMs. Additionally, concurrent work Search-r1 [21] and R1-Searcher [43] both use answer-based reward to improve the reasoning in RAG. MAO-ARAG [5] proposes a hierarchical RAG framework that achieves a balance between effectiveness and efficiency

in RAG by dynamically orchestrating executors through optimizing the planner agent with RL, thereby achieving Pareto optimality. SoftRankPO [52] introduces a meta-cognitive multi-agent reinforcement learning framework that enables LLM agents to dynamically deliberate, coordinate, and improve reasoning performance. Additionally, [28] propose an AI search paradigm which can be trained with RL algorithms.

Some other works use DPO [33] or similar alignment algorithms to optimize LLMs. A noise-filtering method [62] is proposed by optimizing mutual information between compressed data and output while minimizing it with the retrieved passage. ATM [61], an Adversarial Tuning Multi-agent system, enhances the robustness and performance of retrieval-augmented generators in question answering by iteratively tuning against an adversarial attacker agent to better discriminate useful documents and resist fabricated content. SEER [59] proposes a novel self-aligned evidence extraction learning framework aimed at enhancing RAG performance by optimizing the extraction of high-quality, concise, and relevant evidence. RAG-DDR [27] optimizes RAG systems by aligning data preferences between modules through DDR, resulting in enhanced performance on knowledge-intensive tasks.

## B   How to Construct Training Data for the SFT Training Process

Before the multi-module joint training process of MAPPO, the warm-up to each trainable modules is necessary. The warm-up can make LLMs to follow the instructions better and reduce the exploration space for the MAPPO training process to stabilize and accelerate the joint training process.

In this section, we introduce the details of constructing training data for each modules.

### B.1   Query Rewriter

Query Rewriter is to reformulate the initial query $q$, which may be too complex or ambiguous to resolve with a single retrieval, into a set of sub-questions denoted as $subq$. The prompt of Query Rewriter is in Table 3.

In Rewrite-Retrieve-Read [30], a small language model was trained using PPO to effectively rewrite queries for RAG. Building on this approach, we utilize the publicly available query rewriting data from Rewrite-Retrieve-Read as the SFT dataset to warm start the Query Rewriter in MMOA-RAG.

Table 3: The prompt of Query Rewriter agent.

---

**system:** You are a professional assistant skilled at rewriting complex or unclear questions into simpler, more searchable subquestions.

**assistant:** Okay, I will provide the rewritten sub-questions.

**user:** Please help me rewrite or decompose the given questions into sub-questions, making them easier to search for answers in a search engine. The rewritten sub-questions must have logical connections and dependencies, without being overly repetitive in meaning. Additionally, avoid using vague demonstrative pronouns and similar terms.

**assistant:** Okay, I will provide the rewritten sub-questions.

**user:** Original question is {content of Question}. Now rewrite or decompose the original question into sub-questions according to the above requirements, and only output the rewritten subquestions in the format of one subproblem per line without any additional content. Additionally, avoid using vague demonstrative pronouns and similar terms, avoid the duplicate subquestions.

---

## B.2 Selector

The task of the Selector is to choose a subset $D_\text{selected}$ that are helpful for answering a question from a given set $D$ with $K$ candidate documents.

The output format of the Selector is the IDs of the documents in $D_\text{selected}$ (e.g., Document0, Document4, Document6, Document7), as shown in the prompt of Selector in Table 4. Therefore, to construct SFT data for the Selector, the ground truth should be the IDs of documents that are truly useful for answering the question. One method to obtain the ground truth is to employ advanced LLMs, such as GPT-4o, to provide the ground truth. However, we have found that this approach does not yield results as good as expected. Additionally, BGM [22] introduced and optimized a bridge module which is similar to the Selector module. They proposed a method called synthesis silver passage sequence (Synthesis SPS) to construct the ground truth for SFT data. However, the Synthesis SPS method requires examining each candidate document $d_{i,j} \in D_i$ (candidate documents of question $i$), invoking the LLM for each check, and comparing the utility values before and after the check, making it a complex and costly method.

Table 4: The prompt of Selector agent.

> **system:** You are a helpful, respectful and honest assistant. Your task is to output the IDs of the candidate Documents (0,1,2,...,K-1) which are helpful in answering the Question.
>
> **assistant:** Okay, I will provide the ID of candidate Documents which are helpful in answering the Question.
>
> **user:** Question: {content of Question}
> Document0: {content of Document0}
> Document1: {content of Document1}
> ⋮
> Document(K-2): {content of Document(K-2)}
> Document(K-1): {content of Document(K-1)}
>
> **assistant:** OK, I received the Question and the candidate Documents.
>
> **user:** Now, output the IDs of the candidate Documents (0,1,2,...,K-1) which are helpful in answering the Question: {content of Question}, for example, in the following format: Document0,Document4,Document6,Document7.

We propose a convenient heuristic approach for constructing SFT data, aimed at LLMs to effectively follow instructions and output in the desired format. As illustrated in Figure 4, for a given question $q_i$ and its golden answer, there are $K$ candidate documents denoted as $d_{i,j}$, where $j \in \{0, 1, \cdots, K-1\}$. First, by removing certain insignificant stop words and punctuation marks from $q_i$ and its golden answer, and converting the words to lowercase, we obtain the set $Set_{q_i}$. Similarly, we perform the same operation on the $K$ candidate documents $d_{i,j}$ to obtain $Set_{d_{i,j}}$. Finally, if any word from $Set_{q_i}$ appears in $Set_{d_{i,j}}$, the ID of corresponding document $j$ is included in the final output as the Label of SFT. With this approach, we can rapidly and cost-effectively construct the Selector's ground truth labels during the SFT stage. Given our focus on the subsequent joint optimization of multiple modules, this straightforward data construction method can adequately meet our requirements.

## B.3 Generator

The Generator is responsible for producing the final answer, $Ans_\text{predict}$, based on the $D_\text{selected}$ provided by the Selector. Therefore, the ground truth for the SFT data of Generator is the golden answer $Ans_\text{golden}$. And the prompt of Generator is in Table 5.

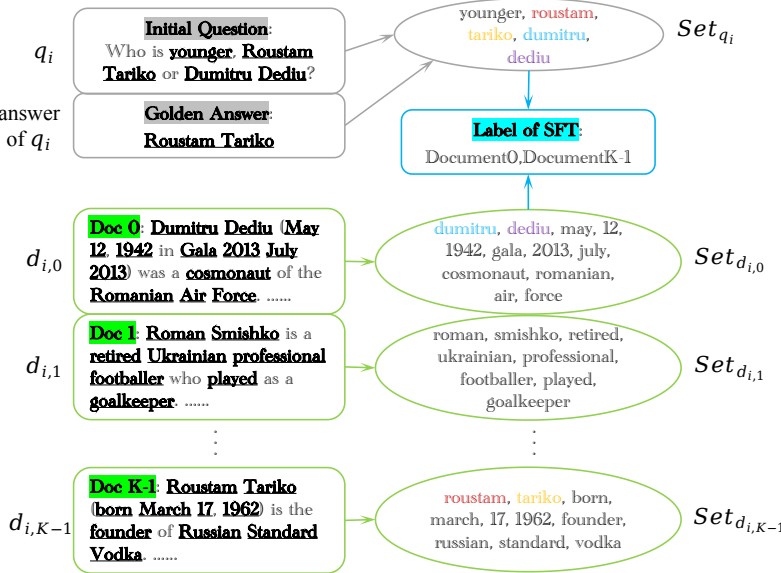

Figure 4: The convenient approach to construct the SFT data for Selector.

Table 5: The prompt of Generator agent.

> **system:** You are a helpful, respectful and honest assistant. Your task is to predict the answer to the question based on the given documents. If you don't know the answer to a question, please don't share false information. Answer the question as accurately as possible.
>
> **assistant:** Okay, I will provide the answer to the question based on the corresponding documents. Please provide the question and the corresponding documents.
>
> **user:** Question: {content of Question}
> Document0: {content of Document0}
> Document3: {content of Document3}
> ⋮
> Document7: {content of Document7}
>
> **assistant:** OK, I received the Question and the corresponding Documents.
>
> **user:** Given the Question and the corresponding Documents, predict the answer to the Question as briefly and accurately as possible based on the Documents. Only give the brief and accurate answer with the form of **answer** and nothing else.

With these approaches, the SFT data used for the warm start training of these three modules—Query Rewriter, Selector, and Generator—can be obtained. All modules can be fine-tuned using the typical loss function of SFT presented in Equation (17).

$$\mathcal{L}_{\text{SFT}}(\theta) = -\sum_{n=1}^{N} \log P(Y_i \mid X_i; \theta) \tag{17}$$

In Equation (17), $N$ represents the number of samples in the SFT dataset, while $\theta$ denotes the parameters of the LLM. The variable $X_i$ corresponds to the input content of each module. Meanwhile, $Y_i$ signifies the output content of each module.

## C    The Pseudocode of Multi-Agent Training Process of MMOA-RAG

**Algorithm 1** is the pseudocode for multi-agent optimization based on MAPPO.

---

**Algorithm 1:** The Training Process of Multi-Agent Optimization

---

**Initialize**: The parameters of the Actor model $\theta$, the Critic model $\phi$, the SFT model $\theta_{\text{SFT}}$, and a replay buffer $\mathcal{M} = \varnothing$.

**Inputs**: Dataset with initial questions $q$ and corresponding golden answers $Ans_{\text{golden}}$

**for** $epoch \leftarrow 1$ **to** $N\_epoch$ **do**

  **for** $batch \leftarrow 1$ **to** $N\_batch$ **do**

    // Collect Rollout

    **for** *each question* $q \in batch$ **do**

      // Query Rewriter (QR)

      Construct observation $O_{QR}$ according to Equation (1)

      Get sub-questions $subq$ for initial question $q$

      Calculate the penalty term of Query Rewriter $P_{QR}$

      // Retriever

      Retrieve $K$ candidate documents to construct $D$

      // Selector (S)

      Construct observation $O_S$ according to Equation (4)

      Select $IDs$ of helpful documents, and get $D_{\text{selected}}$

      Calculate the penalty term of Selector $P_S$

      // Generator (G)

      Construct observation $O_G$ according to Equation (7)

      Predict the $Ans_{\text{predict}}$ to initial question $q$

      Calculate the penalty term of Generator $P_G$

      // Getting Reward and Storing Tuple

      Calculate the F1 score of $Ans_{\text{predict}}$ as the shared reward $R_{\text{shared}}$

      Get reward for each agent $R_i, i \in \{QR, S, G\}$, according Equation (3), (6) and (9)

      Store tuple $\mathcal{T} = ((O_{QR}, subq, R_{QR}), (O_S, IDs, R_S), (O_G, Ans_{\text{predict}}, R_G))$ in the replay buffer $\mathcal{M}$

    // Policy and Value Optimization

    **for** *each question* $q \in batch$ **do**

      Compute the advantage function $\hat{A}_{\pi_\theta}^{i,t}$ using GAE

      Calculate the loss of the Actor $\mathcal{L}_{\text{Actor}}(\theta)$ and Critic model $\mathcal{L}_{\text{Critic}}(\phi)$

      Update the parameters of models through the overall loss function $\mathcal{L}(\theta, \phi)$ in Equation (10)

  Clear the replay buffer $\mathcal{M}$ to $\varnothing$

**Output :** Well-trained Actor model with parameters $\theta_{\text{trained}}$

---

## D    The Detailed Introduction of the Implementation Details

### D.1    Baselines

The methods detailed below serve as baseline models:

**LLM w/o RAG**: This approach answers questions solely based on the internal knowledge embedded within LLMs, without employing any retrieval mechanisms.

**Vanilla RAG w/o train**: This method leverages a retrieval model to obtain relevant documents, thereby augmenting the LLM's internal knowledge with external sources. Here, the LLM remains in a pre-trained state and has not undergone any fine-tuning.

**Vanilla RAG w SFT**: Building on the Vanilla RAG framework, this variant involves a fine-tuned LLM to improve the integration of retrieved external knowledge with the LLM's internal knowledge, potentially enhancing the quality of final answers.

**SELF-RAG** [1]: This innovative framework advances LLM performance by incorporating both adaptive retrieval mechanisms and self-reflection processes, aiming to produce precise and dependable answers.

**RetRobust** [54]: This approach fortifies the RAG architecture against irrelevant contexts, thereby boosting its effectiveness in open-domain question-answering scenarios.

**Rewrite-Retrieve-Read** [30]: A small-scale query rewriter model is trained using reinforcement learning, optimizing the interaction between retrieval and answer generation.

**BGM** [22]: Utilizing PPO, this method trains a bridge component to filter and identify documents that are more likely to be helpful, thus refining the quality of the retrieved context.

**RAG-DDR** [27]: This approach utilize DPO to align data preferences between different modules, enhancing performance and reducing hallucinations in RAG.

## D.2 Implementation Details

To ensure fairness in comparison, all baselines were re-implemented using Llama-3-8B-Instruct as the backbone architecture. Within these methods, the untuned modules employed the pre-trained version of Llama-3-8B-Instruct, whereas the trainable modules were derived through specific SFT processes on Llama-3-8B-Instruct. Notably, in the Rewrite-Retrieve-Read framework, the query rewrite module, which is trainable, was optimized using the PPO algorithm applied to Llama-3-8B-Instruct; meanwhile, answer generation was implemented based on the SFT-refined backbone. Regarding the BGM method, we rebuilt the bridge to connect the retrieval model and the generation model, leveraging Llama-3-8B-Instruct, with this bridge being trained using the PPO algorithm. The generation model for BGM was similarly obtained from the SFT-refined backbone.

We utilize Contriever [16] as the Retriever. Regardless of how many sub-questions $subq$ the Query Rewriter generates from the initial question $q$, the Selector consistently receives a fixed set of $K = 10$ documents as input. For example, if the Query Rewriter yields 2 sub-questions, each sub-question is used for retrieval, with the top-5 documents from each retrieval being selected as part of the candidate documents $D$ for the Selector. Furthermore, it is important to emphasize that we do not utilize any support facts or positive passages [4] that come with the official datasets to generate answers. Instead, we only use the $K$ candidate documents from the retrieval model as external knowledge for answer generation.

Besides, we also employ Llama-3-8B-Instruct [7] as the foundational LLM for the baselines and MMOA-RAG. Building on the PPO code from LLama-Factory[5] [60], we have developed MMOA-RAG, which optimizes the RAG multi-agent system using Multi-Agent PPO. And the critical hyperparameters of MMOA-RAG are detailed in Table 6.

Table 6: Key hyperparameters in the training process of MMOA-RAG.

| Name | Explanation | Values |
|---|---|---|
| $\beta_{max}$ | Maximum $\beta$ in Equation (15) | 0.2 |
| $\beta_{min}$ | Minimum $\beta$ in Equation (15) | 0.06 |
| $\gamma$ | Key hyperparameter in GAE | 1.0 |
| $\lambda$ | Key hyperparameter in GAE | 0.95 |
| $\epsilon$ | Clip range in MAPPO | 0.2 |
| $\alpha$ | Coefficients in Equation (10) | 0.1 |
| $lr$ | Maximum learning rate | 2e-5 |
| `bueffer_size` | Buffer size in MAPPO | 128 |
| `lr_scheduler` | Learning rate scheduler | cosine |
| `top_p` | Sampling parameters in training | 0.9 |

---

[4]Some QA datasets inherently include supportive texts that aid in answering questions. And some studies incorporate these supportive texts alongside retrieved candidate documents as input to LLMs for answer prediction, which significantly enhances answer quality. However, we adhere to the natural RAG process by using only the candidate documents provided by the retriever for answer prediction, as annotated supportive texts are not present in the natural RAG workflow.

[5]`https://github.com/hiyouga/LLaMA-Factory`

Table 7: The performance of different methods based on Retriever BGE [47] and E5 [49]. Llama-3-8B-Instruct and its the fine-tuned version is the backbone for all methods. The highest baseline value in each dataset is underscored. The Δ displays the improvement of MMOA-RAG over the best baseline.

| Methods | HotpotQA | | | 2WikiMultihopQA | | | AmbigQA | | |
|---|---|---|---|---|---|---|---|---|---|
| | Acc | EM | F1 | Acc | EM | F1 | Acc | EM | F1 |
| **Retriever: BGE [47]** | | | | | | | | | |
| Vanilla RAG w/o train | 37.72 | 28.99 | 41.62 | 28.86 | 18.54 | 27.87 | 48.05 | 35.10 | 52.31 |
| Vanilla RAG w SFT | 45.47 | 41.16 | 54.32 | 42.79 | 41.50 | 46.70 | 47.10 | 42.45 | 57.41 |
| Self-RAG [1] | 33.41 | 30.59 | 42.12 | 37.39 | 36.41 | 40.91 | 30.00 | 27.00 | 40.77 |
| RetRobust [54] | 46.66 | 42.74 | 55.77 | 47.03 | **45.57** | 50.62 | 47.00 | 42.20 | 58.14 |
| Rewrite-Retrieve-Read [30] | 45.58 | 40.95 | 54.35 | 43.91 | 42.48 | 47.83 | 46.30 | 41.75 | 57.19 |
| BGM [22] | 45.50 | 41.49 | 54.73 | 41.88 | 40.67 | 45.53 | 48.15 | 42.70 | 58.61 |
| RAG-DDR [27] | 45.13 | 41.51 | 54.32 | 42.49 | 41.21 | 46.00 | 48.50 | 42.95 | 58.71 |
| MMOA-RAG | **47.22** | **43.46** | **56.45** | **47.14** | 45.47 | **50.94** | **48.65** | **43.45** | **59.10** |
| Δ | **+0.56** | **+0.72** | **+0.68** | **+0.11** | -0.10 | **+0.32** | **+0.15** | **+0.50** | **+0.39** |
| **Retriever: E5 [49]** | | | | | | | | | |
| Vanilla RAG w/o train | 36.92 | 28.01 | 40.80 | 28.57 | 17.50 | 27.67 | **50.50** | 37.00 | 53.93 |
| Vanilla RAG w SFT | 46.48 | 42.08 | 55.51 | 44.74 | 43.32 | 48.66 | 48.20 | 43.40 | 58.53 |
| Self-RAG [1] | 33.64 | 30.84 | 42.36 | 38.39 | 37.38 | 42.03 | 30.10 | 27.20 | 40.83 |
| RetRobust [54] | 46.66 | 42.78 | 55.85 | 49.27 | **47.59** | 52.75 | 48.85 | 44.10 | 59.29 |
| Rewrite-Retrieve-Read [30] | 46.62 | 42.00 | 55.54 | 46.41 | 44.83 | 50.35 | 47.85 | 43.55 | 58.36 |
| BGM [22] | 45.81 | 41.72 | 55.32 | 43.24 | 41.99 | 46.83 | 48.35 | 44.05 | 59.04 |
| RAG-DDR [27] | 44.50 | 41.09 | 53.96 | 43.89 | 42.49 | 47.29 | 49.60 | 43.75 | 59.75 |
| MMOA-RAG | **47.46** | **43.73** | **57.23** | **49.32** | 47.53 | **53.13** | 50.45 | **44.80** | **60.80** |
| Δ | **+0.80** | **+0.95** | **+1.38** | **+0.05** | -0.06 | **+0.38** | -0.05 | **+0.70** | **+1.05** |

# E  The Performance of Different Methods Based on Retriever BGE and E5

We firstly perform the training of different methods based on retrieval model Contriever [16], and the results are shown in Table 1.

We further test the trained models on different retriever BGE [47] and E5 [49]. From Table 7, we can see that our MMOA-RAG also surpasses most of baselines on each datasets.

# F  Out-of-Domain Experiments

We also conducted out-of-domain (OOD) experiments to evaluate the generalization capabilities of MMOA-RAG compared to the baselines. We trained LLM on HotpotQA dataset and evaluated on the AmbigQA dataset. The experimental results are shown in Table 8.

Table 8: Out-of-domain experimental results: The model is trained on the HotpotQA dataset and tested on the AmbigQA dataset.

| Methods | Acc | EM | F1 |
|---|---|---|---|
| SELF-RAG [1] | 26.70 | 24.25 | 36.38 |
| RetRobust [54] | 34.19 | 31.75 | 44.08 |
| Rewrite-Retrieve-Read [30] | 33.91 | 30.73 | 43.61 |
| BGM [22] | 32.58 | 29.77 | 42.07 |
| MMOA-RAG (ours) | **35.45** | **32.43** | **45.62** |

From Table 8, it is evident that MMOA-RAG demonstrates superior performance in the OOD experiments, underscoring its notable generalization capabilities and effectively. Additionally, it is noteworthy that RetRobust outperforms all other baselines. This can be attributed to its strategy of integrating both relevant and irrelevant data during the SFT process, which significantly enhances the robustness and generalization abilities of the RAG system.

# G   Discussion about Training Time and Inference Time

**Training Time**    In our MMOA-RAG framework, multiple agents (modules) are optimized simultaneously, which results in increased training time as the number of agents rises. When we have $n$ agents, the time required for rollouts and updating model parameters becomes $n$ times that of a single agent. However, since this increase in training time is linear relative to the number of agents, it does not lead to an excessively large overhead. Therefore, this is entirely manageable.

**Inference Time**    The inference process of MMOA-RAG follows the workflow outlined in Figure 1. Training with multiple agents does not introduce any additional overhead during the inference phase.

# H   Reasons for Using Reward Sharing and Parameter Sharing Strategies

## H.1   Reward sharing

It is through the way of reward sharing that the optimization objectives of multiple modules can be unified and aligned to optimize the quality of the final predicted answer. Multiple modules in MMOA-RAG naturally fit into a fully cooperative relationship, and reward sharing is a near-standard setting in fully cooperative MARL algorithms [34, 29, 55, 3], as fully cooperative modeling only allows all agents to have the same goal (here, F1 score). Therefore, reward sharing is reasonable and necessary here.

## H.2   Parameter sharing

Parameter sharing is also a common and effective strategy in multi-agent reinforcement learning [34, 29, 55, 3]. Parameter sharing means that multiple agents share the parameters of the same policy network instead of maintaining a separate policy network for each agent. The biggest advantage of this approach is that it can reduce the computational and storage overhead: sharing parameters can significantly reduce the total number of parameters of the model, thus reducing the computational complexity and storage requirements. This is particularly important for RAG systems with multiple modules based on LLM, which is an important reason why we chose to use the parameter sharing strategy.

However, it is undeniable that parameter sharing does have its limitations: it can lead to instability when the number of agents increases. However, in our framework, there are three agents (modules) that are jointly optimized, and the problem of instability generally does not occur when the number of agents is small. The experimental results also prove that under the setting of parameter sharing, the functions of Query Rewriter, Selctor and Generator are not affected at all, and they can complete the task well and exceed the baselines without parameter sharing such as RRR and BGM.

To summarize, we adopt the Settings of parameter sharing and reward sharing commonly used in previous MARL frameworks, which are common and versatile multi-agent reinforcement learning frameworks. Agents are distinguished by different prompts, which correspond to different observations of different agents. Therefore, there doesn't have to be a distinction at the architectural or parameter level to be called different agents.

# I   Case Study

In this section, we analyze the question from the HotpotQA dataset: "The Vermont Catamounts men's soccer team currently competes in a conference that was formerly known as what from 1988 to 1996?" The golden answer to this question is "the North Atlantic Conference." We compare two inference cases, Inference Case 1 involves the response process of MMOA-RAG after MAPPO training, while Inference Case 2 pertains to the response process before MAPPO training, where only SFT was conducted:

- In Inference Case 1, we observe that the LLM reformulates the initial question $q$ into two sub-questions $subq$. In Candidate Documents $D$, Document 0 to 4 are retrieved based on Sub-question 1, and Documents 5 to 9 are retrieved based on Sub-question 2. Notably, the answer highlighted in red appears in Document 5, which the Selector accurately identifies. The

final answer generated based on the Initial question $q$ and Document 5 is "North Atlantic Conference," consistent with the Golden answer $Ans_{\text{golden}}$. This process effectively demonstrates how, after MAPPO training, multiple modules can collaborate accurately to provide the most precise answer possible.

- In contrast, Inference Case 2 shows that the LLM only reformulates the initial question $q$ into a single Sub-question 1. All 10 candidate documents are retrieved based on this Sub-question 1. Upon inspection, none of these documents contain text that can answer the initial question. The Selector also chooses multiple documents, which clearly does not aid in generating the correct answer. Consequently, the predicted answer $Ans_{\text{predict}}$ is the incorrect "Yankee Conference." Hence, prior to MAPPO training, despite each module being well-optimized through SFT, there is a lack of effective collaboration among the modules, potentially leading to inaccurate responses.

These examples provide an intuitive understanding of the advantages of joint multi-module optimization within the complex RAG framework.

---

**Inference Case 1: Training with MAPPO**

- **Initial question** $q$:
  - The Vermont Catamounts men's soccer team currently competes in a conference that was formerly known as what from 1988 to 1996?
- **Golden answer** $Ans_{\text{golden}}$:
  - the North Atlantic Conference
- **Sub-questions** $subq$:
  - Sub-question 1: name of the conference that the Vermont Catamounts men's soccer team currently competes in
  - Sub-question 2: name of the conference from 1988 to 1996
- **Candidate Documents** $D$:
  - **Document 0:**
    * Title: Vermont Catamounts football
    * Content: The Vermont Catamounts football program were the intercollegiate American football team for the University of Vermont located in Burlington, Vermont. The team competed in the NCAA Division I and were members of the Yankee Conference. The school's first football team was fielded in 1886. The football program was discontinued at the conclusion of the 1974 season. Vermont fields a team at the club football level, in a conference that also uses the Yankee Conference name. The first intercollegiate game in the state of Vermont happened on November 6, 1886 between Dartmouth and Vermont in Burlington. Dartmouth won
  - **Document 1:**
    * Title: Vermont Catamounts men's ice hockey
    * Content: The Vermont Catamounts men's ice hockey team is a National Collegiate Athletic Association (NCAA) Division I college ice hockey program that represents the University of Vermont. The Catamounts are a member of Hockey East, joining in 2005 after competing in ECAC Hockey from 1974-2005. They play home games at Gutterson Fieldhouse in Burlington, Vermont. Vermont has appeared in the NCAA Men's Hockey Championship five times since making the move to Division I in 1974-75 including trips to the Frozen Four in 1996 and 2009. Prior to moving to Division I, UVM competed in ECAC Division
  - **Document 2:**
    * Title: Vermont Catamounts football
    * Content: Members of the conference. Vermont began Yankee Conference play in 1947 with Connecticut, Maine, Massachusetts, New Hampshire, and Rhode Island. Although they played UMass and UNH in the first season, they didn't play Maine until 1950, Rhode Island until 1955, and UConn until 1966. Boston University began league play in 1973. Notable alumni include: Vermont Catamounts football The Vermont Catamounts football program were the intercollegiate American football team for the University of Vermont located in Burlington, Vermont. The team competed in the NCAA Division I and were members of the Yankee Conference. The school's first football team was fielded in
  - **Document 3:**
    * Title: Vermont Catamounts men's basketball
    * Content: The Vermont Catamounts Basketball team is the basketball team that represents the University of Vermont in Burlington, Vermont. The school's team currently competes in the America East Conference and plays its home games at Patrick Gym. The team has reached the NCAA Division I Men's Basketball Tournament six times, in 2003, 2004, 2005, 2010, 2012, and 2017. UVM famously upset Syracuse University in the first round of the 2005 tournament. The Catamounts are coached by John Becker. America East Coach of the Year America East Player of the Year America East Defensive Player of the Year
  - **Document 4:**
    * Title: Vermont Catamounts
    * Content: The Vermont Catamounts are the varsity intercollegiate athletic programs of the University of Vermont, based in Burlington, Vermont, United States. The school sponsors 18 athletic programs (8 men's, 10 women's), most of which compete in the NCAA Division I America East Conference (AEC), of which the school has been a member since 1979. The men's and women's ice hockey programs compete in Hockey East. The men's and women's alpine and nordic skiing teams compete in the Eastern Intercollegiate Ski Association (EISA). The school's athletic director is Robert Corran. The Catamounts have won six national championships, all in skiing.

- **Document 5:**
  * Title: America East Conference
  * Content: The America East Conference is a collegiate athletic conference affiliated with the NCAA Division I, whose members are located mainly in the Northeastern United States, specifically New England. Its nine members include the public flagship universities of three states, and one private university. The America East Conference was founded as the Eastern College Athletic Conference-North, a men's basketball-only athletic conference in 1979. The conference was known as the Eastern College Athletic Conference-North from 1979 to 1988 and **the North Atlantic Conference from 1988 to 1996**. The charter members were the University of Rhode Island, the College of

- **Document 6:**
  * Title: Northwest Conference
  * Content: The Northwest Conference (NWC) is an athletic conference which competes in the NCAA's Division III. Member teams are located in the states of Oregon and Washington. The NWC was formed in 1926, making it one of the oldest athletics conferences in the western United States. For 60 years, the Northwest Conference sponsored sports exclusively for men, but in 1984 it joined with the Women's Conference of Independent Colleges to become the Northwest Conference of Independent Colleges, shortening the name to its current moniker in 1996 when it joined the NCAA. The charter members included Willamette University, Pacific University,

- **Document 7:**
  * Title: Big West Conference
  * Content: The Big West Conference (BWC) is an American collegiate athletic conference whose member institutions participate in the National Collegiate Athletic Association's Division I. The conference was originally formed in 1969 as the Pacific Coast Athletic Association (PCAA) and in 1988 was renamed the Big West Conference. The conference stopped sponsoring college football after the 2000 season. The Big West Conference was formed in June 1968 as the Pacific Coast Athletic Association. The five original charter members were Fresno State, San Jose State, UC Santa Barbara, San Diego State, and Long Beach State. Two other schools, Cal State

- **Document 8:**
  * Title: Big Eight Conference Men's Basketball Tournament
  * Content: The Big Eight Conference was an NCAA athletic conference that existed from 1907 to 1996, when it and the Southwest Conference disbanded to create the Big 12 Conference. The post-season conference tournament was instituted in 1977 and from that time the winner won the conference's automatic NCAA Tournament bid. From 1977 until 1985, the quarterfinals were played on the campus sites of the higher seeded teams. The last Big Eight men's basketball tournament ran through the conference's final season in 1996. Missouri won the most Big Eight tournament titles with six. Colorado was the

- **Document 9:**
  * Title: Big South Conference Men's Basketball Tournament
  * Content: The Big South Conference Men's Basketball Tournament (popularly known as the Big South Tournament) is the conference championship tournament in basketball for the Big South Conference. The tournament has been held every year since 1986. It is a single-elimination tournament and seeding is based on regular season records. The winner, declared conference champion, receives the conference's automatic bid to the NCAA men's basketball tournament. However, the conference did not have an automatic bid to the NCAA tournament from 1986 to 1990. From 2003 through 2012, the tournament was held mostly at campus sites. In

- **IDs of Selected Documents** $D_{\text{selected}}$:
  - Document 5
- **Predicted answer** $Ans_{\text{predict}}$:
  - North Atlantic Conference ✓

---

## Inference Case 2: Training with SFT

- **Initial question** $q$:
  - The Vermont Catamounts men's soccer team currently competes in a conference that was formerly known as what from 1988 to 1996?
- **Golden answer** $Ans_{\text{golden}}$:
  - the North Atlantic Conference
- **Sub-questions** $subq$:
  - Sub-question 1: name of the conference that the Vermont Catamounts men's soccer team currently competes in
- **Candidate Documents** $D$:
  - **Document 0:**
    * Title: Vermont Catamounts football
    * Content: Vermont Catamounts football The Vermont Catamounts football program were the intercollegiate American football team for the University of Vermont located in Burlington, Vermont. The team competed in the NCAA Division I and were members of the Yankee Conference. The school's first football team was fielded in 1886. The football program was discontinued at the conclusion of the 1974 season. Vermont fields a team at the club football level, in a conference that also uses the Yankee Conference name. The first intercollegiate game in the state of Vermont happened on November 6, 1886 between Dartmouth and Vermont in Burlington. Dartmouth won
  - **Document 1:**

- * Title: Vermont Catamounts men's ice hockey
- * Content: Vermont Catamounts men's ice hockey The Vermont Catamounts men's ice hockey team is a National Collegiate Athletic Association (NCAA) Division I college ice hockey program that represents the University of Vermont. The Catamounts are a member of Hockey East, joining in 2005 after competing in ECAC Hockey from 1974-2005. They play home games at Gutterson Fieldhouse in Burlington, Vermont. Vermont has appeared in the NCAA Men's Hockey Championship five times since making the move to Division I in 1974-75 including trips to the Frozen Four in 1996 and 2009. Prior to moving to Division I, UVM competed in ECAC Division

- **Document 2:**
  - * Title: Vermont Catamounts men's basketball
  - * Content: Vermont Catamounts men's basketball The Vermont Catamounts Basketball team is the basketball team that represents the University of Vermont in Burlington, Vermont. The school's team currently competes in the America East Conference and plays its home games at Patrick Gym. The team has reached the NCAA Division I Men's Basketball Tournament six times, in 2003, 2004, 2005, 2010, 2012, and 2017. UVM famously upset Syracuse University in the first round of the 2005 tournament. The Catamounts are coached by John Becker. America East Coach of the Year America East Player of the Year America East Defensive Player of the Year

- **Document 3:**
  - * Title: Vermont Catamounts football
  - * Content: members of the conference. Vermont began Yankee Conference play in 1947 with Connecticut, Maine, Massachusetts, New Hampshire, and Rhode Island. Although they played UMass and UNH in the first season, they didn't play Maine until 1950, Rhode Island until 1955, and UConn until 1966. Boston University began league play in 1973. Notable alumni include: Vermont Catamounts football The Vermont Catamounts football program were the intercollegiate American football team for the University of Vermont located in Burlington, Vermont. The team competed in the NCAA Division I and were members of the Yankee Conference. The school's first football team was fielded in

- **Document 4:**
  - * Title: Vermont Catamounts
  - * Content: Vermont Catamounts The Vermont Catamounts are the varsity intercollegiate athletic programs of the University of Vermont, based in Burlington, Vermont, United States. The school sponsors 18 athletic programs (8 men's, 10 women's), most of which compete in the NCAA Division I America East Conference (AEC), of which the school has been a member since 1979. The men's and women's ice hockey programs compete in Hockey East. The men's and women's alpine and nordic skiing teams compete in the Eastern Intercollegiate Ski Association (EISA). The school's athletic director is Robert Corran. The Catamounts have won six national championships, all in skiing.

- **Document 5:**
  - * Title: 2018–19 Vermont Catamounts men's basketball team
  - * Content: to win their conference tournament, the Catamounts received an automatic bid to the National Invitation Tournament, where they lost to Middle Tennessee in the First Round. !colspan=12 style=| Exhibition !colspan=12 style=| Non-conference regular season !colspan=9 style=| America East Conference regular season !colspan=12 style=| America East Tournament Source 2018–19 Vermont Catamounts men's basketball team The 2018–19 Vermont Catamounts men's basketball team will represent the University of Vermont in the 2018–19 NCAA Division I men's basketball season. They will play their home games at the Patrick Gym in Burlington, Vermont and will be led by 8th-year head coach John Becker. The Catamounts

- **Document 6:**
  - * Title: Vermont
  - * Content: based in Burlington. They were named the Vermont Expos before 2006. Up until the 2011 season, they were the affiliate of the Washington Nationals (formerly the Montreal Expos). Currently the highest teams in basketball, representing Vermont are the NCAA's Vermont Catamounts – male and female. The Vermont Frost Heaves, the 2007 and 2008 American Basketball Association national champions, were a franchise of the Premier Basketball League, and were based in Barre and Burlington from the fall of 2006 through the winter of 2011. The Vermont Bucks, an indoor football team, were based in Burlington and began play in 2017 as

- **Document 7:**
  - * Title: 2017–18 Vermont Catamounts men's basketball team
  - * Content: 2017–18 Vermont Catamounts men's basketball team The 2017–18 Vermont Catamounts men's basketball team represented the University of Vermont during the 2017–18 NCAA Division I men's basketball season. The Catamounts, led by seventh-year head coach John Becker, played their home games at Patrick Gym in Burlington, Vermont as members of the America East Conference. They finished the season 27–8, 15–1 in America East play to win the America East regular season championship. The Catamounts defeated Maine and Stony Brook to advance to the championship game of the America East Tournament where they lost to UMBC. As a regular season conference champion

- **Document 8:**
  - * Title: Vermont Catamounts
  - * Content: basketball team has performed well in the America East Tournament. In 2013, it made a conference-record 18th appearance in the semifinals. It has the most wins in tournament play, with 35 (2013 field). It has advanced at least one round in 19 of the 24 tournaments. The Catamounts were the first women's basketball program to go undefeated during the regular season in back-to-back seasons (1991-1992 and 1992-1993), a feat matched only by Connecticut (2008-2009 and 2009-2010). In the 2012-2013 season, the program had a total attendance mark of 10,579. After the 2009 season, the Vermont baseball program, which played at

- **Document 9:**
  - * Title: 2018–19 Vermont Catamounts men's basketball team
  - * Content: 2018–19 Vermont Catamounts men's basketball team The 2018–19 Vermont Catamounts men's basketball team will represent the University of Vermont in the 2018–19 NCAA Division I men's basketball season. They will play their home games at the Patrick Gym in Burlington, Vermont and will be led by

8th-year head coach John Becker. The Catamounts finished the 2017–18 season 27–8, 15–1 in the America East Conference play to finish in first place. In the America East Tournament, they defeated Maine and Stony Brook to advance to the championship game, where they lost to UMBC. As a regular season conference champion who failed

- **IDs of Selected Documents** $D_{\text{selected}}$:
  - Document 0, Document 2, Document 3, Document 4, Document 5, Document 6, Document 8, Document 9
- **Predicted answer** $Ans_{\text{predict}}$:
  - Yankee Conference ✗

