# OpenReview forum: "Improving Retrieval-Augmented Generation through Multi-Agent Reinforcement Learning"
_NeurIPS.cc/2025/Conference — NeurIPS 2025 poster_

### Official Review · Reviewer_zfzg · 2025-06-20

**Clarity:** 3
**Significance:** 2
**Originality:** 3
**Rating:** 5
**Confidence:** 3

**Summary:**

In this paper the authors present a way of treating a RAG pipeline as a multi-agent collaborative task, where each component (query rewriting, document selection, answer generation) becomes a RL agent. Through multi-agent reinforcement learning, they are harmonised to operate toward the same goal (e.g. improving F1 score) and show its usefulness in a series of evaluations (examining also the benefit of combining all three components together).

**Questions:**

Are the penalties described in section 3.3 really needed? Eg. the one about not writing too many sub-questions (with the hardcoded value 4), what happens if they are not applied an have you tested other values?

**Ethical Concerns:**

["NO or VERY MINOR ethics concerns only"]

**Final Justification:**

The authors have addressed my main feedback showing how our method applies beyond Q&A to other domains

**Quality:**

3

**Strengths And Weaknesses:**

I have overall enjoyed reading this paper and I found the idea of combining multi-agents in a cooperative task to perform RAG via reinforcement learning insightful. I would suggest the authors to give more details about the computational infrastructure used and the overall length/cost of training at the end of Section 3, as I believe readers would be interested to know more about that step.

The main weakness I have noticed is about the choice of evaluation, as I have found the datasets picked very limited to only QA settings for a method that should in theory be very impactful across many RAG applications. I would strongly recommend the authors to consider benchmarking their approach in other settings where RAG would play a role beyond QA, for instance decision-support (e.g. medical retrieval), summarisation, etc.

---

> ### Author Rebuttal · Authors · 2025-07-29
>
> First of all, thank you for your review. Here is our response:
> # Weakness 1
> The main weakness I have noticed is about the choice of evaluation, as I have found the datasets picked very limited to only QA settings for a method that should in theory be very impactful across many RAG applications. I would strongly recommend the authors to consider benchmarking their approach in other settings where RAG would play a role beyond QA, for instance decision-support (e.g. medical retrieval), summarisation, etc.
>
> # Response to W1
> Despite most existing RAG work being tested on QA datasets, your suggestion to test in other scenarios is valuable.
>
> Therefore, we tested the F1 score of different methods on the **LOFin**, **a financial decision-making dataset**. And we also test methods on **PubMed**, **a novel biomedical decision-making dataset**. Additionally, we conducted tests based on **a regularly updated dataset**, the **FreshQA**, which is updated weekly with new versions to incorporate potential new test data.
>
> As seen in the table below, MMOA-RAG also demonstrates an advantage over baselines in decision-making and latest questions.
>
> | Method                | LOFin | PubMed | FreshQA |
> |-----------------------|----------------|---------|----------------|
> | Vanilla RAG w SFT     | 12.23          | 37.50    | 17.78          |
> | Self-RAG              | 14.08          | 50.5   | 16.44          |
> | Rewrite-Retrieve-Read | 14.91          | 38.00    | 17.94          |
> | BGM                   | 10.41          | 39.5   | 17.97          |
> | MMOA-RAG              | **15.02**          | **55.5**  | **18.82**          |
>
> # Question 1
> Are the penalties described in section 3.3 really needed? Eg. the one about not writing too many sub-questions (with the hardcoded value 4), what happens if they are not applied an have you tested other values?
>
> # Response to Q1:
> **Firstly, it is clear that penalty terms are necessary.**
>
> **Why design penalty terms**: During the RL training process, the agent continuously explores new state and action spaces during rollout and utilizes the obtained data. If an agent’s output becomes repetitive or deviates from the expected paradigm during exploration and exploitation, penalty terms are needed to constrain the agent’s exploration scope. This helps prevent an agent’s output from crashing, which could adversely affect the integrated optimization of the entire RAG system’s multiple modules.
>
> **What happens if penalty terms are removed**: Initially, in MARL training, no anomalies will be observed. However, after a certain step, an agent’s output might not meet expected requirements, potentially causing the entire training process to crash (e.g., a significant drop in reward). For instance, after a certain step, the Query Rewriter might output redundant and repetitive sub-questions, harming the training process. Similarly, after a particular step, the Selector might repeatedly output the IDs of candidate documents, thus losing its function of selecting useful documents. The Generator may, as exploration progresses, tend to output overly long and ambiguous answers. If any agent experiences a breakdown and is not penalized promptly, it can lead to a rapid decline in the overall performance of the RAG system.
>
> **Have I tested values other than 4**: Yes, I have tested other values. The purpose of setting this threshold is to prevent the Query Rewriter from outputting redundant or repetitive sub-questions. Through case studies, it is known that almost all questions can be decomposed into four or fewer sub-questions. Therefore, I tried N=4, 5, and 6. These values can all prevent the Query Rewriter from exhibiting crashing behavior during MARL training. In other words, the hyperparameters in these penalty terms are not sensitive, as long as they serve the purpose of constraining the agent’s exploration space to prevent it from going out of control.
>
> Here is our complete response to your questions, and I hope it addresses your concerns. I look forward to your reply.

---

> > ### Comment · Reviewer_zfzg · 2025-08-04
> >
> > Thank you for your answer and for having addressed all my suggestions - I'm happy to increase my overall score and please do include the evaluation you ran for addressing W1 to your appendix as it would be useful to other researchers

---

> > > ### Author Response · Authors · 2025-08-05
> > >
> > > We appreciate your response and are pleased to have addressed your concerns.

---

### Official Review · Reviewer_xDLX · 2025-06-20

**Clarity:** 2
**Significance:** 2
**Originality:** 2
**Rating:** 4
**Confidence:** 3

**Summary:**

This paper presents MMOA-RAG, a Multi-Module Joint Optimization Algorithm for Retrieval-Augmented Generation (RAG) systems. The authors model the RAG pipeline—consisting of a query rewriter, document selector, and answer generator—as a cooperative multi-agent reinforcement learning (Co-MARL) framework. Each module is treated as an agent optimized via Multi-Agent Proximal Policy Optimization (MAPPO), with a shared reward based on the final answer’s F1 score. The approach is evaluated on three open-domain QA datasets (HotpotQA, 2WikiMultihopQA, AmbigQA), demonstrating improved performance over several baselines.

**Questions:**

1. Can you clarify the distinct motivation for each experiment and what unique insight each offers?
2. How sensitive is the final performance to the choice of these penalties? Are they critical for convergence or only auxiliary?

**Ethical Concerns:**

["NO or VERY MINOR ethics concerns only"]

**Final Justification:**

Response to W1:
The authors reiterate their motivation but ultimately rely on a direct application of MAPPO, without any notable algorithmic adaptation to RAG-specific challenges. Modeling RAG modules as cooperative agents is conceptually reasonable but not novel, and without tailored techniques to handle the unique structure, dependencies, or reasoning patterns in RAG systems, the methodological contribution remains limited.

Response to W3:
Format violations during RL fine-tuning are a well-understood and solvable issue, especially with proper initialization from SFT checkpoints, as seen in models like DeepSeek. The reliance on a penalty term to maintain format integrity appears ad hoc and unnecessary, addressing a minor technical hurdle rather than a core challenge. The justification overstates the importance of this design choice.

Response to W4:
The claim that specialization eliminates the risk of forgetting overlooks the real concern: highly specialized agents likely require distinct parameter adaptations. Sharing a single LLM backbone across them introduces a clear risk of functional interference. The authors provide no empirical analysis or ablation to evaluate this, nor consider alternatives like partial sharing, which weakens the rationale behind their chosen setup.

**Limitations:**

Consider mentioning compute cost, inference latency, or implications for real-time deployment of large RAG systems. Being transparent about such limitations will not be penalized and can improve the paper’s credibility.

**Paper Formatting Concerns:**

No obvious formatting issues.

**Quality:**

3

**Strengths And Weaknesses:**

Strengths:
1. End-to-End Optimization: The method directly optimizes downstream answer quality, addressing the common misalignment between retriever and generator objectives.
2. The formulation of RAG modules as cooperative agents within a MARL framework is conceptually neat and modular. The design of observations, actions, and rewards for each agent is well-specified and logically structured.

Weakness:
1. The core method is largely a direct adaptation of MAPPO to the RAG context, with minimal innovation tailored to the unique challenges of RAG systems.
2. The related work is not sufficiently comprehensive and lacks a clear positioning of the proposed method relative to existing RAG optimization frameworks.
3. Although the paper defines penalty terms to regularize agent behavior, their effectiveness is not empirically validated.
4. All modules share the same underlying LLM parameters, but the paper lacks any discussion or experiment addressing potential issues like functional interference or catastrophic forgetting.
5. The generalization (Section 4.3) and ablation (Section 4.4) experiments are overly similar, both aiming to show the benefits of joint optimization, leading to redundancy.

---

> ### Author Rebuttal · Authors · 2025-07-25
>
> First of all, thank you for your review. Here is our response:
>
> # Weakness 1
>
> The core method is largely a direct adaptation of MAPPO to the RAG context, with minimal innovation tailored to the unique challenges of RAG systems.
>
> # Response to W1
>
> **We have indeed addressed one of the unique challenges inherent in RAG systems.**
>
> Let us reiterate the research motivation of our paper (which is clearly stated in the Abstract): current RAG systems are largely modular, consisting of multiple modules. However, the optimization objectives of these modules are not aligned with the ultimate goal of generating high-quality answers. Therefore, we modeled each module within the RAG system as an agent and conceptualized the modular RAG system as a fully cooperative multi-agent scenario. We used the Multi-Agent RL algorithm MAPPO for optimization. Through this approach, we can unify and align the optimization objectives of all modules within the RAG system towards generating high-quality answers.
>
> Thus, **the primary issue we address is the unification and alignment of optimization objectives across multiple modules within RAG systems.**
>
> # Weakness 2
>
> The related work is not sufficiently comprehensive and lacks a clear positioning of the proposed method relative to existing RAG optimization frameworks.
>
> # Response to W2
>
> Due to space constraints in the main text, it is difficult for us to provide a comprehensive overview of related works in Section 2 "Related Work". We have mentioned in the end of Section 2 that “**More detailed related works can be seen in Appendix A.**”
>
> Appendix A provides a detailed explanation of the related works. We divided RAG into two major categories: RAG without Parameters Updating and RAG with Parameters Updating. The latter is further divided into Optimizing RAG with SFT and Optimizing RAG with RL. Since we use a MARL algorithm to optimize the RAG system, **our approach falls under the Optimizing RAG with RL category.**
>
> # Weakness 3
>
> Although the paper defines penalty terms to regularize agent behavior, their effectiveness is not empirically validated.
>
> # Response to W3
>
> **The purpose of designing a penalty term for each agent is to prevent any agent from producing outputs that do not meet the requirements during the MARL training process, thereby avoiding disruptions in the joint optimization of multiple modules.**
>
> Initially, we did not include a penalty term, and each agent’s reward was based solely on the F1 score of the final generated answer. Since RL agents need to continuously explore new state and action spaces, this exploration can lead to situations where an agent’s output does not conform to the required format, which in turn can halt the entire joint training process. However, after introducing the penalty term, when an agent explores outputs that do not meet the requirements, the penalty term can penalize such behavior, allowing the entire training process to continue running smoothly and stably.
>
> In summary, **the inclusion of the penalty term transformed the multi-module MARL joint training system from being prone to failure to operating stably, which is the best evidence of the penalty term’s effectiveness.**
>
> **In addition, due to the rebuttal policy prohibiting the display of any images, I will describe the difference between using and not using penalty terms**: We conducted training on HotpotQA for a total of 600 steps. At around 300 steps, the test F1 score for both with and without penalty terms was approximately 0.475. However, around the 370-step mark, the Query Rewriter without penalty terms began outputting repeated sub-questions, which directly led to the collapse of multi-module training. At 400 steps, the test results showed that the F1 score with penalty terms was close to 0.48, whereas without penalty terms, due to the training collapse, the F1 score was below 0.2. This is the difference in using penalty terms. **We will include related experiments and analysis in future versions of the paper.**
>
> # Weakness 4
>
> All modules share the same underlying LLM parameters, but the paper lacks any discussion or experiment addressing potential issues like functional interference or catastrophic forgetting.
>
> # Response to W4
>
> **Parameter sharing is a very common setting in cooperative multi-agent reinforcement learning [1,2,3,4]**. When the number of agents, N, is relatively small, it can ensure that the performance among agents is almost unaffected while significantly reducing the number of parameters and computational overhead (reducing the parameters from O(N) to O(1)). This is why we adopt parameter sharing. Moreover, the experimental results in Table 1 and Table 7 demonstrate that our method achieves better performance across multiple datasets, proving the effectiveness of parameter sharing and indicating that there is no interference between the functions of different agents.
>
> Regarding the issue of catastrophic forgetting: In the RAG scenario, the functions of different modules are highly customized. For example, the Query Rewriter only needs to focus on decomposing and rewriting the question, while the Generator only needs to provide accurate answers based on the candidate text. **Therefore, in a modular RAG system, the function of each agent needs to be highly specialized rather than comprehensive and general-purpose, making the issue of catastrophic forgetting inapplicable in a modular RAG scenario.**
>
> [1] QMIX: Monotonic Value Function Factorisation for Deep Multi-Agent Reinforcement Learning
>
> [2] Multi-Agent Actor-Critic for Mixed Cooperative-Competitive Environments
>
> [3] The Surprising Effectiveness of PPO in Cooperative, Multi-Agent Games
>
> [4] PTDE: Personalized Training with Distilled Execution for Multi-Agent Reinforcement Learning
>
> # Weakness 5
>
> The generalization (Section 4.3) and ablation (Section 4.4) experiments are overly similar, both aiming to show the benefits of joint optimization, leading to redundancy.
>
> # Response to W5
>
> In the paper, we explicitly states that **Section 4.3** and **Section 4.4** correspond to **RQ2** and **RQ3**, respectively:
>
> **RQ2**: How does the joint optimization of individual modules in the RAG pipeline contribute to the effectiveness of the MMOA-RAG framework? **RQ3**: Can MMOA-RAG exhibit generalizability across different RAG systems?
>
> Specifically, **in Section 4.3**, the pipeline for each RAG method is the same, consisting of agents i in {Query Rewriter (QR), Retriever (R), Selector (S), and Generator (G)}. “MMOA-RAG w/o i” means freezing agent i during MARL joint optimization. Therefore, Section 4.3 investigates whether the performance declines when only two of the three modules are optimized during MARL joint optimization.
>
> In contrast, **in Table 2 and Section 4.4**, the three pipelines are different. The QR+S+G pipeline consists of QR, R, S, and G. The S+G pipeline comprises R, S, and G, while the QR+G pipeline includes QR, R, and G. These represent three different modular RAG systems, and all LLM agents undergo MARL joint optimization. Here, we aim to study whether there is a significant improvement in the effectiveness of these differently composed RAG systems before and after MARL joint optimization. This would demonstrate that our MMOA-RAG approach can be applied to optimize various RAG systems (generalizability).
>
> **The research questions we aim to verify in Section 4.3 and Section 4.4 are different, so there is no issue of experimental redundancy.**
>
> # Question 1
>
> Can you clarify the distinct motivation for each experiment and what unique insight each offers?
>
> # Response to Q1
>
> Please see the **Response to Weakness 5**.
>
> # Question 2
>
> How sensitive is the final performance to the choice of these penalties? Are they critical for convergence or only auxiliary?
>
> # Response to Q2
>
> Please see the **Response to Weakness 3**.
>
> # Limitations:
> Consider mentioning compute cost, inference latency, or implications for real-time deployment of large RAG systems. Being transparent about such limitations will not be penalized and can improve the paper’s credibility.
>
> # Response to Limitations
> **We have discussed issues regarding training and inference time in Appendix G.**
>
> Additionally, regarding the limitations of the paper, **We will further consider the limitations of our algorithm from both the algorithmic and systems perspectives and update them in the next version of the paper.** For example, I will focus on the following points:
> (1) Our method’s pipeline is fixed, and we need to explore the effectiveness of multi-module joint optimization in a more general and dynamic RAG system.
> (2) The penalty term is a crucial component for maintaining training stability, but its design relies on experience and cannot fully rely on the F1 score to update the entire system.
>
> Here is our complete response to your questions, and I hope it addresses your concerns. I look forward to your reply.

---

> > ### Comment · Reviewer_xDLX · 2025-08-03
> >
> > Response to W1:
> > The authors reiterate their motivation but ultimately rely on a direct application of MAPPO, without any notable algorithmic adaptation to RAG-specific challenges. Modeling RAG modules as cooperative agents is conceptually reasonable but not novel, and without tailored techniques to handle the unique structure, dependencies, or reasoning patterns in RAG systems, the methodological contribution remains limited.
> >
> > Response to W3:
> > Format violations during RL fine-tuning are a well-understood and solvable issue, especially with proper initialization from SFT checkpoints, as seen in models like DeepSeek. The reliance on a penalty term to maintain format integrity appears ad hoc and unnecessary, addressing a minor technical hurdle rather than a core challenge. The justification overstates the importance of this design choice.
> >
> > Response to W4:
> > The claim that specialization eliminates the risk of forgetting overlooks the real concern: highly specialized agents likely require distinct parameter adaptations. Sharing a single LLM backbone across them introduces a clear risk of functional interference. The authors provide no empirical analysis or ablation to evaluate this, nor consider alternatives like partial sharing, which weakens the rationale behind their chosen setup.

---

> > > ### Author Response · Authors · 2025-08-05
> > >
> > > We have provided further clarification and responses to your concerns.
> > >
> > > In particular, we have elaborated on the penalty design (**W3**) and compared our parametric sharing method with the non-parametric sharing method (**W4**) through experimental approaches.
> > >
> > > We hope this addresses your concerns effectively and look forward to receiving your further response.

---

> ### Author Response · Authors · 2025-08-03
>
> Thanks for you response. Here is our further response.
>
> # Regarding the statement “rely on a direct application of MAPPO, without any notable algorithmic adaptation to RAG-specific challenges”
> Our main contributions are as follows:
>
> (1) We observed that existing RAG optimization methods overlook the misalignment of objectives among multiple modules within the RAG system, which affects the generation of high-quality answers. Therefore, we employed MARL algorithms to unify and optimize the objectives of all modules.
>
> (2) In this process, we designed penalty terms for each module to stabilize training and used an outcome-based reward (F1 score) as a global reward to optimize the complex RAG system. **This is the special aspect we designed for the RAG system**.
>
> (3) **We are the first to model the modular RAG system as a fully cooperative problem of multiple agents (modules), which has not been done in previous work.**
>
> (4) Using the MAPPO algorithm to optimize the multi-module RAG system allows the modules to learn cooperative relationships, as shown in our appendix’s Inference Case Study. **This precisely addresses the “dependencies” you mentioned in the RAG system**.
>
> # Regarding the statement “overstates the importance of this design choice (penalty terms for each agent)”
> In the fine-tuning of LLMs with around 8B parameters, formatting issues are not easily resolved. **Even if the model is initially warm started by SFT, as the RL training process progresses, the agent will continue to explore new state and action spaces, and the output format of the model of this size will still have problems.** During RL optimization of models (Referring to models around 8B rather than DeepSeek level 671B), output format collapse, repetition, or entropy collapse are common issues.
>
> **More importantly, in a multi-agent reinforcement learning scenario, if one agent’s output is problematic, it affects the training of the entire system**. The table below presents the F1 score results before and after the removal of the penalty term for the Query Rewriter from the overall reward. As evident from the table, eliminating the penalty term causes the training system to collapse, making the joint optimization of the RAG system unattainable. **Thus, penalty terms are essential and maintain the stability of the joint optimization of the whole RAG system. If the penalty term is removed, it is likely to lead to the collapse of the training.**
>
> **This also addresses your last concern.** Due to the issues inherent in a multi-module RAG system, necessary penalty terms are a special design we implemented for the MARL algorithm in the joint optimization of a modular RAG system. In other words, we are not merely applying MAPPO to RAG without consideration.
>
> | Steps        | 0 step    | 350 step   | 370 step  | 400 step   |
> |--------------|-------|-------|------------------------------|-------|
> | MMOA-RAG with penalty | 0.447 | 0.477 |  | 0.480 |
> | MMOA-RAG w/o penalty  | 0.447 | 0.472 | collapse of multi-module training | 0.183 |
>
> # Regarding “Sharing a single LLM backbone across them introduces a clear risk of functional interference”
> Following your suggestion, we also experimented with a non-parameter sharing approach, equipping each agent with an adapter. From the table below, we can see that **the effects of parameter sharing or not are similar**. This demonstrates that in our multi-module RAG scenario, **parameter sharing does not lead to functional interference among modules**. Thus, **our parameter sharing approach is reasonable, ensuring that different agents’ functions do not interfere while reducing training costs.** This is also consistent with the conclusions of previous works [1,2,3] on MARL in reinforcement learning simulation environments such as Starcraft II.
>
> | Dataset            | Parameter Sharing | Non-Parameter Sharing |
> |-|-|-|
> | HotpotQA           | 36.15             | 36.19            |
> | 2WikiMultihopQA    | 41.52             | 41.39            |
> | AmbigQA            | 34.75             | 34.66            |
>
> [1] QMIX: Monotonic Value Function Factorisation for Deep Multi-Agent Reinforcement Learning
>
> [2] The Surprising Effectiveness of PPO in Cooperative, Multi-Agent Games
>
> [3] PTDE: Personalized Training with Distilled Execution for Multi-Agent Reinforcement Learning
>
> Here is our further response and experiments, which we hope will address your concerns.

---

> > ### Comment · Reviewer_xDLX · 2025-08-06
> >
> > My concerns have been clearly resolved. I hope the author will include these experiments and innovative points in the article so that more readers can learn about your work.

---

> > > ### Author Response · Authors · 2025-08-06
> > >
> > > We appreciate your response and are pleased to have addressed your concerns. We will supplement the paper according to your comments.

---

### Official Review · Reviewer_Vxtd · 2025-07-02

**Clarity:** 3
**Significance:** 3
**Originality:** 2
**Rating:** 4
**Confidence:** 4

**Summary:**

This paper proposes MMOA-RAG, a multi-agent reinforcement learning (MARL) framework to optimize multiple modules within a Retrieval-Augmented Generation (RAG) pipeline—namely the Query Rewriter, Selector, and Generator—by modeling them as cooperating agents optimized using a shared reward (F1 score). The paper claims improvements over prior work by extending end-to-end optimization to more components and aligning their objectives via a unified reward using MAPPO.
While the general motivation—addressing misalignment among modules in RAG—is sound and well-motivated, the core contributions and empirical validations raise several concerns regarding novelty, methodological clarity, and evaluation rigor.

**Questions:**

Have you considered training each agent with separate LLM heads (e.g., adapter tuning) to allow for more expressive role specialization?

**Ethical Concerns:**

["NO or VERY MINOR ethics concerns only"]

**Final Justification:**

This is a solid piece of work. The authors have conducted numerous additional validation experiments during the rebuttal phase in response to the requests, which justifies a positive score.

**Limitations:**

see the weakness

**Quality:**

3

**Strengths And Weaknesses:**

Strengths

1. Well-motivated problem: The paper tackles an important and known issue in RAG pipelines—disjoint training of components—and seeks to unify the optimization process.

2. Systematic framework: The authors clearly formalize each RAG component as an agent and describe how shared rewards are used to encourage cooperation.

Weaknesses

1. Limited Novelty Beyond Extended Scope: While the authors position the work as an end-to-end optimization framework, this line of work has already been well explored in prior literature ([1], [2]). The only substantial difference here is the inclusion of additional modules (Query Rewriter and Selector) into the training loop. This extension, while practical, is relatively incremental, and the contribution is overstated as novel multi-agent collaboration.

2. All “agents” share the same underlying LLM, differentiated only through prompts. There is no architectural or parameter-level separation between agents, which makes the claim of multi-agent learning weak. Without decoupled policies or distinct learning dynamics, the formulation seems closer to a multi-task learning setup with shared parameters and shared reward, rather than a genuine multi-agent RL framework.

3. The ablation analysis is underdeveloped. The current results only test the impact of removing modules from joint optimization, which is insufficient to support the claim of agent-level collaboration. Specifically: There is no analysis on how each module performs when trained independently and then frozen. There is no comparison across different LLMs or analysis of whether shared vs. non-shared parameterization impacts performance.

---

> ### Author Rebuttal · Authors · 2025-07-25
>
> First of all, thank you for your review. Here is our response:
>
> # Weakness 1
>
> Limited Novelty Beyond Extended Scope: While the authors position the work as an end-to-end optimization framework, this line of work has already been well explored in prior literature ([1], [2]). The only substantial difference here is the inclusion of additional modules (Query Rewriter and Selector) into the training loop. This extension, while practical, is relatively incremental, and the contribution is overstated as novel multi-agent collaboration.
>
> # Response 1 to W1
>
> First of all, there may be some problems. I did not see the "prior literature ([1], [2])" you mentioned in your review.
>
> Furthermore, we are extending end-to-end optimization framework to more modules (including Query Rewriter, Selector, Generator), which means that we align the optimization goals of multiple modules across the RAG pipeline with the ultimate unified goal of generating high-quality answers. This greatly alleviates the problem that the goals of multiple modules in RAG are inconsistent with the actual optimization goals, and our experimental results in Table 1 and Table 7 prove that our approach can indeed improve the effect of RAG. **This is our main contribution, which is not a simple extension of existing work.**
>
> And we model the entire RAG pipeline as a cooperative multi-agent reinforcement learning (Co-MARL) framework and optimize it with MAPPO algorithm. **Our modeling way and optimization method are never covered by previous works on RAG, so in essence our method is significantly different from the previous works.**
>
> # Weakness 2
>
> All “agents” share the same underlying LLM, differentiated only through prompts. There is no architectural or parameter-level separation between agents, which makes the claim of multi-agent learning weak. Without decoupled policies or distinct learning dynamics, the formulation seems closer to a multi-task learning setup with shared parameters and shared reward, rather than a genuine multi-agent RL framework.
>
> # Reponse 2 to W2
>
> ### Reward sharing:
>
> For the reward sharing you mentioned, this is one of the core of this paper. It is through the way of reward sharing that the optimization objectives of multiple modules can be unified and aligned to optimize the quality of the final predicted answer. Multiple modules in MMOA-RAG naturally fit into a fully cooperative relationship, and **reward sharing is a near-standard setting in fully cooperative MARL algorithms [1,2,3,4], as fully cooperative modeling only allows all agents to have the same goal (here, F1 score). Therefore, reward sharing is reasonable and necessary here.**
>
> ### Parameter sharing:
>
> **Parameter sharing is also a common [1,2,3,4] and effective strategy in multi-agent reinforcement learning.** Parameter sharing means that multiple agents share the parameters of the same policy network instead of maintaining a separate policy network for each agent. The biggest advantage of this approach is that it can reduce the computational and storage overhead: sharing parameters can significantly reduce the total number of parameters of the model, thus reducing the computational complexity and storage requirements. This is particularly important for RAG systems with multiple modules based on LLM, which is an important reason why we chose to use the parameter sharing strategy.
>
> However, it is undeniable that parameter sharing does have its limitations: it can lead to instability when the number of agents increases. However, in our framework, there are three agents (modules) that are jointly optimized, and the problem of instability generally does not occur when the number of agents is small. The experimental results also prove that under the setting of parameter sharing, the functions of Query Rewriter, Selctor and Generator are not affected at all, and they can complete the task well and exceed the baselines without parameter sharing such as RRR and BGM.
>
> **To summarize, we adopt the Settings of parameter sharing and reward sharing commonly used in previous MARL frameworks, which are common and versatile multi-agent reinforcement learning frameworks.** Agents are distinguished by different prompts, which correspond to different observations of different agents. **Therefore, there doesn't have to be a distinction at the architectural or parameter level to be called different agents.**
>
> [1] QMIX: Monotonic Value Function Factorisation for Deep Multi-Agent Reinforcement Learning
>
> [2] Multi-Agent Actor-Critic for Mixed Cooperative-Competitive Environments
>
> [3] The Surprising Effectiveness of PPO in Cooperative, Multi-Agent Games
>
> [4] PTDE: Personalized Training with Distilled Execution for Multi-Agent Reinforcement Learning
>
> # Weakness 3
>
> The ablation analysis is underdeveloped. The current results only test the impact of removing modules from joint optimization, which is insufficient to support the claim of agent-level collaboration. Specifically: There is no analysis on how each module performs when trained independently and then frozen. There is no comparison across different LLMs or analysis of whether shared vs. non-shared parameterization impacts performance.
>
> # Response 3 to W3
>
> The current results only test the impact of removing modules from joint optimization" you mentioned corresponds to Section 4.4, And **that's not the only kind of ablation experiments we did.**
>
> In addition, the "how each module performs when trained independently and then frozen" mentioned by you is carried out **in Section 4.3, We did perform ablation experiments by freezing each module separately during MARL training**. It can be seen from Figure 2 that the effect of freezing a certain module is worse than jointly optimizing all modules, which proves the advantage and necessity of jointly optimizing all modules.
>
> As for "There is no comparison across different LLMs" you mentioned, it is true that we only fine-tuned the model based on llama3-8b-instruct, which has some damage to prove the generality of our method. However, the experiments in Section 4.4 demonstrate the generality of our approach across different workflows. And the focus of our verification is whether the performance of the RAG system can be steadily improved after the joint optimization of MARL.
>
> **Analysis of whether shared vs. non-shared parameterization impacts performance**: In practice, parameter sharing is widely accepted in multi-agent reinforcement learning (MARL) because it significantly reduces the computational overhead associated with maintaining independent parameters, while incurring only a minimal loss in performance. Thus, the impact of non-shared parameters on performance is quite small (compared to shared parameters), and good performance can still be achieved with parameter sharing. Our experimental results were obtained by training on a machine with 8\*A800 GPUs, whereas using a non-shared parameter approach would require 24\*A800 GPUs to achieve the same batch size during training. Therefore, adopting parameter sharing is an ideal solution in LLM-based multi-module RAG scenarios. **And we will add the analysis of whether shared vs. non-shared parameterization impacts performance in our paper.**
>
> # Question 1
>
> Have you considered training each agent with separate LLM heads (e.g., adapter tuning) to allow for more expressive role specialization?
>
> # Response to Q1
>
> Yes, thank you very much for your advice. I think fine-tuning for each agent via adapter tuning is really a great approach and I will try your suggestion in the future. But as argued earlier, this does not mean that full parameter sharing is a bad solution.
>
> Here is my complete response to your questions, and I hope it addresses your concerns. I look forward to your reply.

---

> > ### Comment · Reviewer_Vxtd · 2025-08-02
> >
> > Thank you for your detailed response. I appreciate the clarifications provided regarding the use of reward and parameter sharing.
> >
> > However, I still find several concerns unresolved. First, while prompt-based differentiation is a lightweight strategy for enabling modular behavior, the use of fully shared model parameters across all components may limit the expressiveness and specialization of each module. In a framework claiming multi-agent collaboration, this raises concerns about whether distinct agent roles are meaningfully learned.
> >
> > Second, the ablation analysis remains underdeveloped. The current studies mainly evaluate the effect of freezing individual modules during training, but do not examine what happens when a module (e.g., the Query Rewriter) is removed entirely from the system. Such an experiment would provide clearer insight into the actual contribution of each module to end-task performance. Additionally, there is no comparison against a baseline where each module is trained independently with its own objective, without joint optimization. Including such a baseline would better support the claim that the proposed multi-agent optimization framework offers a meaningful improvement over simpler alternatives.
> >
> > Finally, while I acknowledge the scalability constraints associated with separate module tuning (e.g., adapter-based or non-shared parameterization), the absence of such analysis further limits the strength of the current claim regarding the benefits of joint MARL-based optimization.
> >
> > In light of these remaining concerns regarding modular specialization, empirical comparisons, and the scope of novelty over prior joint optimization frameworks, I will maintain my original score.

---

> ### Author Response · Authors · 2025-08-03
>
> Thanks for you response. Here is our further response.
> # Regarding “different agents must have different parameters”
> 1. LLMs are widely recognized for their multi-task capabilities, as their pre-training data include a variety of tasks. Prompt engineering is a crucial technique for applying LLMs to different tasks. Therefore, distinguishing different agent roles using prompts is reasonable.
> 2. Moreover, works like MetaGPT [1] operate in multi-agent collaborative scenarios, where the same LLM is used with different prompts to distinguish between different roles/agents.
> 3. LLM-based agents can have modules like long-term memory, short-term memory, and profiles, but these are distinguished through prompts. It is precisely because of LLMs’ multi-task capabilities that LLM-based multi-agent communities [2, 3] can develop. Thus, using prompts to distinguish different agents is entirely reasonable.
> 4. Finally, we used LoRA for joint MARL optimization of the Query Rewriter, Selector, and Generator, and, as per your suggestion, equipped **each agent with a separate adapter**. From the results (F1 score) in the table below, we can see that **the performance of the three modules with complete parameter sharing is similar to having a separate adapter for each agent.** Therefore, **our parameter sharing strategy is entirely feasible in our RAG scenario, as the functions of different agents can be effectively distinguished through prompts without interfering with each other.**
> | Dataset            | Parameter Sharing | Separate Adapter |
> |-|-|-|
> | HotpotQA           | 36.15             | 36.19            |
> | 2WikiMultihopQA    | 41.52             | 41.39            |
> | AmbigQA            | 34.75             | 34.66            |
>
> # Regarding “lack of ablation study”
> 1. Concerning your mention of “not studying the removal of each module from the system individually,” **we has already presented this ablation study in Section 4.4 and Table 2 of our paper**. We removed the Query Rewriter and Selector separately from the RAG pipeline and provided results both before and after MAPPO joint training. We have shown Table 2 below.
>
> **Modules** | **Training Stage & Delta** | **HotpotQA Acc** | **HotpotQA EM** | **HotpotQA F1** | **2Wiki MultihopQA Acc** | **2Wiki MultihopQA EM** | **2Wiki MultihopQA F1** | **AmbigQA Acc** | **AmbigQA EM** | **AmbigQA F1**
> ---|---|---|---|---|---|---|---|---|---|---
> QR+S+G | SFT | 36.00 | 33.04 | 44.69 | 39.54 | 38.50 | 42.97 | 36.55 | 32.60 | 46.71
> | | MAPPO | 39.15 | 36.15 | 48.29 | 42.73 | 41.52 | 46.40 | 38.85 | 34.75 | 48.59
> | | Δ | **+3.15** | **+3.11** | **+3.60** | **+3.19** | **+3.02** | **+3.43** | **+2.30** | **+2.15** | **+1.88**
> S+G | SFT | 34.25 | 32.18 | 43.14 | 38.93 | 37.97 | 42.40 | 35.85 | 32.35 | 45.82
> | | MAPPO | 38.23 | 34.85 | 47.07 | 41.79 | 40.57 | 45.25 | 37.60 | 33.90 | 47.19
> | | Δ | **+3.98** | **+2.67** | **+3.93** | **+2.86** | **+2.60** | **+2.85** | **+1.75** | **+1.55** | **+1.37**
> QR+G | SFT | 36.76 | 32.78 | 45.00 | 39.15 | 37.89 | 42.91 | 35.50 | 31.50 | 45.31
> | | MAPPO | 38.90 | 35.89 | 47.94 | 42.43 | 41.01 | 46.19 | 37.65 | 33.50 | 47.53
> | | Δ | **+2.14** | **+3.11** | **+2.94** | **+3.28** | **+3.12** | **+3.28** | **+2.15** | **+2.00** | **+2.22**
>
> From this, we can draw two conclusions: (1) The performance after MARL training is worse when any module is removed, indicating that both the Query Rewriter and Selector contribute to the overall RAG system. (2) Whether it’s the RAG pipeline before removal (QR+S+G) or the pipelines with either the Query Rewriter (S+G) or Selector (QR+G) removed, the performance after MAPPO training shows a significant improvement compared to SFT, demonstrating the effectiveness of joint MARL optimization.
> 2. Regarding “there is no comparison against a baseline where each module is trained independently with its own objective, without joint optimization,” **we also has presented this experiment in Section 4.4 and Table 2 of the original paper (reproduced in the table below)**. The experimental results show that the performance of different RAG pipelines improves after joint MARL training. This fully demonstrates the effectiveness of joint MARL optimization.
>
> [1] MetaGPT: Meta Programming for A Multi-Agent Collaborative Framework
>
> [2] Large Language Model based Multi-Agents: A Survey of Progress and Challenges
>
> [3] Multi-Agent Collaboration Mechanisms: A Survey of LLMs
>
> Here is our further response and experiments, which we hope will address your concerns.

---

> > ### Comment · Reviewer_Vxtd · 2025-08-05
> >
> > I'm satisfied with the responses - I don't think there are any outstanding questions. I will raise my rating.

---

> > > ### Author Response · Authors · 2025-08-05
> > >
> > > We appreciate your response and are pleased to have addressed your concerns.

---

### Official Review · Reviewer_DCcd · 2025-07-03

**Clarity:** 4
**Significance:** 3
**Originality:** 3
**Rating:** 4
**Confidence:** 4

**Summary:**

The authors propose MMOA-RAG: Multi-Module joint Optimization Algorithm for RAG, which employs multi-agent RL and formulate  the optimization process as a Cooperative Multi-Agent Reinforcement Learning (Co-MARL) to optimize the 4 sub-components (rewrite, retrieve, select, generate) and demonstrate the effectiveness of the approach on , HotpotQA, 2WikiMultihopQA, and AmbigQA, on Llama-3 8B. The proposed algorithm includes 2 components - a warm start SFT, and then Multi-Agent Optimization.

**Questions:**

1. From the workflow in figure 1 page 3, I wonder what was the thought process to isolate out "selector" with "generator", given both of them can be the same LLM - what is the motivation to separate them out?
2. How did you arrive at 4 as the absolute threshold in "PQR is assigned a value of -0.5 if the number of sub-questions exceeds
four, and it is set to 0 if the number of sub-questions is four or fewer".. While 4 or N can be an arbitrary hyperparameter, what I am interested in is the choice to make it a step as opposed to a continuous loss?
3. This statement in the README of the codebase "Therefore, we deployed the retrieval model on a separate machine using Faiss and leveraged GPU acceleration to ensure fast retrieval results." is not clear to me. Are the authors mentioning that FAISS lookup is also GPU memory bound? I was under the impression in your system, the retrieval system stays on the host (NOT device) up until D_{selected}.

**Ethical Concerns:**

["NO or VERY MINOR ethics concerns only"]

**Limitations:**

Yes, but the authors mention "Justification: We discuss some limitations of our proposed method in Appendix G"
But Appendix G reads "G Discussion about Training Time and Inference Time". I would encourage the authors to consider the system as a whole and think about the limitations from algorithmic, and system perspectives. Surely, the proposed algorithm cannot be with "NO limitations".

**Quality:**

3

**Strengths And Weaknesses:**

[+] The authors propose clear techniques to modify each of the loss functions for each agent (Query re-writer, generator, selector, retriever).

[+] The proposed system MMOA-RAG is simple to implement and can benefit RAG systems across the stack.

[+] The paper is very well written

[-] The evals are the weakest link of the paper. I.e., datasets like HotPotQA and others, which tend to be factual, usually are already baked into the pre-training of LLMs such as Llama - I would strongly encourage the authors to consider evaluating their technique on datasets that are post the model release.

[-] To better understand the benefit of each components, For table 1, what would the table look like with just the SFT from "3.4.1 Warm Start with SFT" part of the training. Given, the Rewrite-Retrieve-Read dataset was used for training, it should be a significant contributor, perhaps?

---

> ### Author Rebuttal · Authors · 2025-07-29
>
> First of all, thank you for your review. Here is our response:
> # Weakness 1
> [-] The evals are the weakest link of the paper. I.e., datasets like HotPotQA and others, which tend to be factual, usually are already baked into the pre-training of LLMs such as Llama - I would strongly encourage the authors to consider evaluating their technique on datasets that are post the model release.
>
> # Response to W1
>
> Thanks for your suggestions, we conducted experiments on a QA dataset that is updated regularly, called **FreshQA** (version of **updated on July 21, 2025**) and a finance decision-making dataset, **LOFin**, **released in 2025**. These dates ensures that the FreshQA data and LOFin were not included in the pre-training data for llama3-8b-instruct. Our retrieval model uses E5, and **the corpus is the same Wikipedia dataset used in the main experiments of the paper**.
>
> | Method                | LOFin | Fresh Dataset |
> |-----------------------|----------------|----------------|
> | Vanilla RAG w SFT     | 12.23          | 17.78          |
> | Self-RAG              | 14.08          | 16.44          |
> | Rewrite-Retrieve-Read | 14.91          | 17.94          |
> | BGM                   | 10.41          | 17.97          |
> | MMOA-RAG              | **15.02**          | **18.82**          |
>
> From the table, we can see that MMOA-RAG still achieves the best performance compared to other baselines on both new datasets. This indicates that MMOA-RAG can still provide relatively the best results for questions that were not seen during the LLM pre-training phase. Since the Wikipedia data used in our main experiment is from the 2018 version (which is the version used in most RAG papers), the overall performance on FreshQA might be relatively low. However, if a more updated version of Wikipedia were used, the performance should likely improve further.
>
> # Weakness 2
> [-] To better understand the benefit of each components, For table 1, what would the table look like with just the SFT from "3.4.1 Warm Start with SFT" part of the training. Given, the Rewrite-Retrieve-Read dataset was used for training, it should be a significant contributor, perhaps?
>
> # Response to W2
> **We present the metrics after training with ‘3.4.1 Warm Start with SFT’ in the table below.**
>
> | **Training Stage & Delta** | **HotpotQA Acc** | **HotpotQA EM** | **HotpotQA F1** | **2WikiMultihopQA Acc** | **2WikiMultihopQA EM** | **2WikiMultihopQA F1** | **AmbigQA Acc** | **AmbigQA EM** | **AmbigQA F1** |
> |----------------------------|------------------|-----------------|-----------------|------------------------|------------------------|------------------------|-----------------|----------------|----------------|
> | SFT                        | 36.00           | 33.04           | 44.69           | 39.54                 | 38.50                  | 42.97                  | 36.55           | 32.60          | 46.71          |
> | MAPPO                          | 39.15           | 36.15           | 48.29           | 42.73                 | 41.52                  | 46.40                  | 38.85           | 34.75          | 48.59          |
> | &Delta;                       | **+3.15**       | **+3.11**       | **+3.60**       | **+3.19**             | **+3.02**              | **+3.43**              | **+2.30**       | **+2.15**      | **+1.88**      |
>
> From the above table, it can be observed that applying SFT to all modules already yields good results, and further improvement (&Delta;) is achieved after multi-module joint optimization with MAPPO.
>
> In order to facilitate comparison, we put the performance of baselines in Table 1 of the original paper below for easy reference.
>
> | **Method**                      | **HotpotQA Acc** | **HotpotQA EM** | **HotpotQA F1** | **2WikiMultihopQA Acc** | **2WikiMultihopQA EM** | **2WikiMultihopQA F1** | **AmbigQA Acc** | **AmbigQA EM** | **AmbigQA F1** |
> |--------------------------------|------------------|-----------------|-----------------|------------------------|------------------------|------------------------|-----------------|----------------|----------------|
> | LLM w/o RAG                    | 25.08            | 21.31           | 31.18           | 27.78                  | 23.68                  | 29.47                  | 27.21           | 20.96          | 33.42          |
> | Vanilla RAG w/o train          | 27.99            | 20.62           | 30.67           | 31.94                  | 13.91                  | 22.84                  | 31.09           | 22.42          | 33.56          |
> | Vanilla RAG w SFT              | 36.18            | 32.30           | 44.49           | 39.47                  | 38.28                  | 43.36                  | 34.41           | 30.74          | 44.36          |
> | SELF-RAG                       | 30.42            | 27.77           | 38.93           | 36.32                  | 35.39                  | 38.86                  | 28.35           | 25.70          | 39.04          |
> | RetRobust                      | 37.69            | 34.60           | 46.49           | 41.02                  | 39.73                  | 44.51                  | 35.13           | 32.37          | 44.78          |
> | Rewrite-Retrieve-Read          | 38.03            | 33.93           | 46.32           | 40.40                  | 39.17                  | 44.17                  | 35.94           | 31.90          | 45.92          |
> | BGM                            | 36.05            | 32.76           | 44.54           | 39.61                  | 38.61                  | 43.29                  | 36.01           | 32.53          | 45.76          |
> | RAG-DDR                        | 35.20            | 32.65           | 44.26           | 40.49                  | 39.45                  | 44.18                  | 36.25           | 32.55          | 45.83          |
>
> In addition, regarding the data from Rewrite-Retrieve-Read, we used the query rewrite data they released, which has been a significant contributor to our work.
>
> # Question 1
> From the workflow in figure 1 page 3, I wonder what was the thought process to isolate out "selector" with "generator", given both of them can be the same LLM - what is the motivation to separate them out?
>
> # Response to Q1
> **The motivation to separate them out**: During the RAG process, the introduction of external knowledge can easily lead to noise or conflicting information. Therefore, a "selector" is explicitly introduced to pick out useful text from the candidate documents and filter out useless and harmful information. This helps the generator produce more accurate answers.
>
> In existing work, **BGM [1] and RAG-DDR [2] also have employed similar selector modules** within the RAG pipeline.
>
> [1] (**ACL 2024**) Bridging the Preference Gap between Retrievers and LLMs
>
> [2] (**ICLR 2025**) RAG-DDR: OPTIMIZING RETRIEVAL-AUGMENTED GENERATION USING DIFFERENTIABLE DATA REWARDS
>
> # Question 2
> How did you arrive at 4 as the absolute threshold in "PQR is assigned a value of -0.5 if the number of sub-questions exceeds four, and it is set to 0 if the number of sub-questions is four or fewer".. While 4 or N can be an arbitrary hyperparameter, what I am interested in is the choice to make it a step as opposed to a continuous loss?
>
> # Response to Q2
> You’re referring to a specific issue in RL training. During the RL training process, the model continuously explores new states and action spaces, which can sometimes lead the Query Rewriter to start rewriting the original question into multiple useless or repetitive sub-questions at a certain point in training. This can be detrimental to the overall training process, so we impose penalties for excessive sub-questions to **stabilize training.**
>
> In most cases, this penalty is set to 0, and only when the number of sub-questions exceeds our set threshold (such as N=4) do we impose a penalty of -0.5. This ensures that the exploration by the Query Rewriter remains within a relatively controlled range. When the number of sub-questions is below this threshold, it can generally be considered as normal exploration by the Query Rewriter, so no penalty should be applied. **Therefore, we opted for a step penalty rather than a continuous one.**
>
> # Qustion 3
> This statement in the README of the codebase "Therefore, we deployed the retrieval model on a separate machine using Faiss and leveraged GPU acceleration to ensure fast retrieval results." is not clear to me. Are the authors mentioning that FAISS lookup is also GPU memory bound? I was under the impression in your system, the retrieval system stays on the host (NOT device) up until D_{selected}.
>
> # Response to Q3
> Thank you for providing such detailed questions. Since our training process involves optimizing the Query Rewriter, the rollout process in RL requires real-time retrieval of new sub-questions.
>
> We have deployed the retriever on another machine using FAISS specifically for retrieval purposes, and we have placed the corpus index on the GPU for acceleration (using faiss.index_cpu_to_gpu). We refer to this as **Machine 1**. Therefore, when performing rollouts on the machine used for RL training (referred to as **Machine 2**), when real-time retrieval is required for a given question, we call the retriever deployed on Machine 1 to quickly retrieve the top-k most relevant documents for the given question. **In other words, Machine 2 sends the question text to Machine 1, and the retriever deployed on Machine 1 retrieves the top-k most relevant documents and sends them back to Machine 2.**
>
> # Response to Limitations:
> Thank you for your suggestions. I will update limitations, from algorithmic and system perspectives, in the next version of the paper. For example, I will focus on the following points: 1. Validation was not performed in the dynamic RAG system.  2. The design of the penalty term requires experience.
>
>
> Here is my complete response to your questions, and I hope it addresses your concerns. I look forward to your reply.

---

> ### Author Response · Authors · 2025-08-05
>
> We have provided further clarification and responses to your concerns.
>
> We hope this addresses your concerns and look forward to receiving your further response.

---

> > ### Comment · Reviewer_DCcd · 2025-08-07
> >
> > Thank your for addressing my concerns. If it does get accepted, I'd encourage the authors to add the the "fresh" datasets to the paper. I could comprehend the results from the "FreshQA" but am not able to find the samples for "LOFin" dataset (unless well established e.g., MMLU - I'd suggest the authors share arxiv/github/HF links to datasets).

---

> > > ### Author Response · Authors · 2025-08-07
> > >
> > > First, we appreciate your response and are glad to address your concerns. We will incorporate additional experimental data into the paper as per your comments.
> > >
> > > Regarding the “LOFin” dataset: Considering that the rebuttal policy does not allow sending URL links, we did not provide the paper corresponding to the dataset earlier. Here, we are additionally providing the arXiv link for the LOFin financial dataset: https://arxiv.org/pdf/2505.20368. The corresponding paper has been accepted for ACL 2025 (Findings).

---

### Comment · Area_Chair_dZRJ · 2025-08-04
**Engage in Author-Reviewer Discussions**

Dear reviewers,

If you haven't done so already, please click the 'Mandatory Acknowledgement' button and actively participate in the rebuttal discussion with the authors after carefully reading all other reviews and the author responses.

Thanks,
AC

---

### Note · Authors · 2025-08-11

Dear Reviewers, AC, and SAC,

We are delighted to see that **all the reviewers acknowledge that we have addressed their concerns comprehensively**. We are committed to further strengthening our paper by incorporating their valuable suggestions.

Specifically, we will include experiments and discussions in the paper on why parameter sharing is a feasible approach in the joint optimization scenario of multi-module RAG in MARL, which also addresses **the concerns of Reviewer Vxtd and Reviewer xDLX**. Additionally, we are considering including experimental results in the paper involving new datasets (absent from LLM pre-training data) and domain-specific datasets (medical and financial datasets), as discussed with **Reviewer DCcd and Reviewer zfzg**.

Due to the score-hiding mechanism, we are unaware of any updated scores. **We respectfully request the reviewers to kindly reconsider the ratings in light of the improvements made**. We also **hope that the AC will take into account our diligent efforts to resolve all issues when making the final decision regarding the acceptance of our paper**.

Thank you once again for your thoughtful consideration.

Authors of Paper 8260

---

### Decision · Program_Chairs · 2025-09-17

**Decision:**

Accept (poster)

**Comment:**

This paper proposes a multi-agent reinforcement learning (MARL) framework to optimize multiple modules, Query Rewriter, Selector, and Generator, in a Retrieval-Augmented Generation (RAG) pipeline. In specific, it treats each module as an RL agent and applies individual SFT for warm-up followed by Multi-Agent PPO (MAPPO) for cooperative MARL with a shared reward. Experimental results with LLaMA-3-8B show that the proposed multi-agent joint optimization improves RAG performances over the SFT baseline.

Overall, the proposed MARL for RAG seems to be technically sound with clear motivation. However, as the reviewers’ comment, the technical novelty is somewhat incremental in that it adopts the existing MARL algorithm without any specific modification. Therefore, the main contribution seems to come from the extensive empirical validation. In this respect, the current analysis and ablaltion studies in the current paper are insufficient as the reviewers concerns. However, most concerns are well addressed by the authors during the rebuttal phase including the effects of penalty rewards, shared parameters, and different tasks. Based on the consensus of all positive ratings between the reviewers, I would recommend the paper to be accepted.

However, further empirical validation is needed for the joint optimization with a unified reward (goal) in MARL. For example, it would be important to include a comparison against the case where each module is separately trained by RL with its own dedicated reward. In addition, experiments on various backbone LLMs would be necessary.